# LRRC8/VRAC anion channels enhance β-cell glucose sensing and insulin secretion

Till Stuhlmann[1,2], Rosa Planells-Cases[1] & Thomas J. Jentsch (ID) [1,3]

Glucose homeostasis depends critically on insulin that is secreted by pancreatic β-cells. Serum glucose, which is directly sensed by β-cells, stimulates depolarization- and $Ca^{2+}$-dependent exocytosis of insulin granules. Here we show that pancreatic islets prominently express LRRC8A and LRRC8D, subunits of volume-regulated VRAC anion channels. Hypotonicity- or glucose-induced β-cell swelling elicits canonical LRRC8A-dependent VRAC currents that depolarize β-cells to an extent that causes electrical excitation. Glucose-induced excitation and $Ca^{2+}$ responses are delayed in onset, but not abolished, in β-cells lacking the essential VRAC subunit LRRC8A. Whereas *Lrrc8a* disruption does not affect tolbutamide- or high-$K^+$-induced insulin secretion from pancreatic islets, it reduces first-phase glucose-induced insulin secretion. Mice lacking VRAC in β-cells have normal resting serum glucose levels but impaired glucose tolerance. We propose that opening of LRRC8/VRAC channels increases glucose sensitivity and insulin secretion of β-cells synergistically with $K_{ATP}$ closure. Neurotransmitter-permeable LRRC8D-containing VRACs might have additional roles in autocrine/paracrine signaling within islets.

[1] Leibniz-Forschungsinstitut für Molekulare Pharmakologie (FMP) and Max-Delbrück-Centrum für Molekulare Medizin (MDC), Robert-Rössle-Strasse 10, 13125 Berlin, Germany. [2] Graduate Program of the Faculty for Biology, Chemistry and Pharmacy, Freie Universität Berlin, 14195 Berlin, Germany. [3] Neurocure Cluster of Excellence, Charité Universitätsmedizin Berlin, 10117 Berlin, Germany. Correspondence and requests for materials should be addressed to T.J.J. (email: jentsch@fmp-berlin.de)

nsulin, the only hormone that lowers blood glucose concentrations, is produced and secreted by pancreatic β-cells that constitute about 75% of the islets of Langerhans. Failure to secrete sufficient amounts of insulin results in diabetes mellitus, a common pathology with serious long-term complications that affect several tissues. A rise in serum glucose cell autonomously stimulates β-cell insulin secretion[1]. Glucose sensing by β-cells involves glucose transporter-mediated cellular uptake of glucose and its conversion to ATP and other metabolites. The rise in ATP inhibits $K_{ATP}$ channels (ATP-sensitive potassium channels) expressed in the plasma membrane of β-cells. Since these channels largely control their resting potential, $K_{ATP}$ closure depolarizes β-cells and thereby opens voltage-dependent $Ca^{2+}$ channels. The resulting rise in cytoplasmic calcium triggers exocytosis of insulin-containing granules[2].

The $K_{ATP}$-dependent mechanism for glucose-stimulated insulin secretion is well established, not least by phenotypes resulting from loss- and gain-of-function mutations in either component (Kir6.2 (encoded by *KCNJ11*) and SUR1 (encoded by *ABCC8*)) of the β-cell $K_{ATP}$ channel[2]. However, there may be additional signal transduction cascades controlling insulin secretion[3], as deduced e.g. from glucose-induced insulin secretion in SUR1 or Kir6.2 knock-out (KO) mice[4–6]. Not only the inhibition of $K^+$ channels, but also the opening of $Cl^-$ channels may depolarize β-cells[3,7–9]. Opening of $Cl^-$ channels leads to a depolarizing $Cl^-$ efflux because β-cells accumulate $Cl^-$ above equilibrium[9–12] using $Na^+K^+2Cl^-$ cotransporters[13,14].

It has been proposed that the volume-regulated anion channel VRAC contributes to insulin secretion by depolarizing β-cells in response to glucose-induced cell swelling[7,15]. VRACs (also known as VSOR or VSOAC) appear to be ubiquitously expressed in vertebrate cells. They mediate volume-activated $I_{Cl,vol}$ anion currents that are crucial for regulatory volume decrease (RVD) after hypotonic cell swelling[16,17]. Swelling of pancreatic β-cells indeed induces $I_{Cl,vol}$-like currents[18], but the reported $Cl^- > I^-$ selectivity of those currents[19,20] differs from canonical VRAC[17]. Since the molecular composition of VRAC has been enigmatic until recently[21], attempts to demonstrate a role of VRAC in β-cells were based on notoriously non-specific inhibitors[22–24] precluding conclusive evidence for an involvement of VRAC in glucose sensing or insulin secretion.

We recently identified LRRC8 heteromers as the molecular correlate of VRAC[25]. LRRC8A is the only essential subunit[25,26], but needs at least one of the other LRRC8 (leucine-rich repeat containing 8) isoforms (LRRC8B,-C,-D or -E) to form volume-regulated plasma membrane channels[25]. The LRRC8 subunit composition determines not only biophysical properties of VRAC such as inactivation and single-channel conductance[25,27,28], but more importantly its substrate specificity[29,30]. For instance, VRACs containing LRRC8A and -D subunits conduct various organic compounds including taurine, neurotransmitters, and other signaling molecules[29,30], suggesting the intriguing possibility that VRAC has a role in paracrine or autocrine signaling.

We now generated a mouse model in which we disrupted the essential VRAC subunit LRRC8A specifically in pancreatic β-cells. β-cell swelling, induced by hypotonicity or glucose, activates canonical, LRRC8A-dependent $I_{Cl,vol}$ currents that depolarize the cell. Glucose-induced intracellular $Ca^{2+}$ responses are markedly delayed, but not abolished, in VRAC-deficient β-cells. First-phase glucose-induced insulin secretion by LRRC8A-deficient islets is reduced in vitro, and impaired glucose tolerance of β-cell-specific *Lrrc8a* KO mice suggests an important modulatory role of VRAC in insulin secretion in vivo.

## Results

**Expression and ablation of LRRC8/VRAC channels in β-cells.** To analyze the role of volume-regulated VRAC anion channels in β-cell function and serum glucose regulation, we generated mice in which the essential VRAC subunit LRRC8A[25,26] was specifically deleted in pancreatic β-cells. *Lrrc8a*^lox/lox mice, in which protein-coding exon 3 is flanked by loxP sites (Supplementary Fig. 1), were crossed with Ins-Cre mice that express the Cre-recombinase under the control of the rat insulin 2 promoter[31]. The resulting *Lrrc8a*^lox/lox;Ins-Cre mice (subsequently named βc-Δ8a mice for simplicity) were viable and lacked an overt phenotype.

Western blot analysis revealed that whole pancreas and purified islets expressed all five VRAC subunits (LRRC8A through LRRC8E) (Fig. 1a). In comparison to whole pancreas, however, islets had very little LRRC8E, but expressed larger amounts of LRRC8A, -C and -D (Fig. 1a). For comparison, human pancreatic islets express all LRRC8 subunits as determined by single-cell transcriptome profiling[32]. Although these data are difficult to compare, human LRRC8E appears less expressed in islets compared to exocrine pancreas as found here for mice, but human LRRC8C or -D expression does not seem to differ greatly between islets and acinar/ductal cells[32]. Since we lacked antibodies suitable for detecting LRRC8 subunits in immunohistochemistry, we used knock-in mice expressing an epitope-tagged version of LRRC8D under its endogenous promoter to explore its presence in islets. LRRC8D, which is crucial for VRAC's ability to transport organic compounds[29,30], was strongly expressed in insulin-containing β-cells (Fig. 1b, top). Immunolabeling of LRRC8D-tdTomato stained not only the plasma membrane, but also prominently the cytoplasm of these cells (Fig. 1b). However, we cannot be sure whether this represents a significant presence of native LRRC8D-containing VRACs in intracellular organelles, or rather may have been caused by the fusion of the large epitope to the LRRC8D carboxy-terminus. LRRC8D was also found, albeit at apparently lower levels, in glucagon-secreting α-cells (Fig. 1b, bottom). Consistent with the large preponderance of exocrine over endocrine tissue, pancreatic expression levels of LRRC8A appeared unchanged between control (*Lrrc8a*^lox/lox) and βc-Δ8a mice (Fig. 1c). By contrast, LRRC8A protein levels were markedly decreased in βc-Δ8a islets (Fig. 1c). The magnitude of this decrease (~65%) was roughly compatible with a complete disruption of *Lrrc8a* in β-cells (constituting ~75% of rodent islets[33]) if all islet cells express similar amounts of LRRC8A. Indeed, lacZ staining of islets from mice expressing β-galactosidase under the control of the *Lrrc8a* promoter (Fig. 1d) suggested that all islet cells express similar levels of LRRC8A.

Islet morphology of adult βc-Δ8a mice appeared normal (Fig. 1e) and immunohistochemistry did not reveal changes in insulin or glucagon expression (Fig. 1f). Western blot of pancreas extracts suggested similar insulin levels (Fig. 1c), a finding confirmed by ELISA analysis (Fig. 1g). β-cell-specific disruption of LRRC8A neither changed total pancreatic (Fig. 1h) nor β-cell mass (Fig. 1i). We neither detected changes in the abundance of Kir6.2 (Fig. 1c), the ion-conducting α-subunit of the pivotal $K_{ATP}$ channel, nor did we detect signs for inflammation or increased activated caspase in βc-Δ8a pancreas (Supplementary Fig. 2). Hence neither developmental or degenerative changes, nor indications for altered $K_{ATP}$-dependent glucose signaling were detected in βc-Δ8a pancreas. Moreover, βc-Δ8a mice showed no increased lethality during our observation period of 30 weeks and their weight developed normally during that time (Fig. 1j).

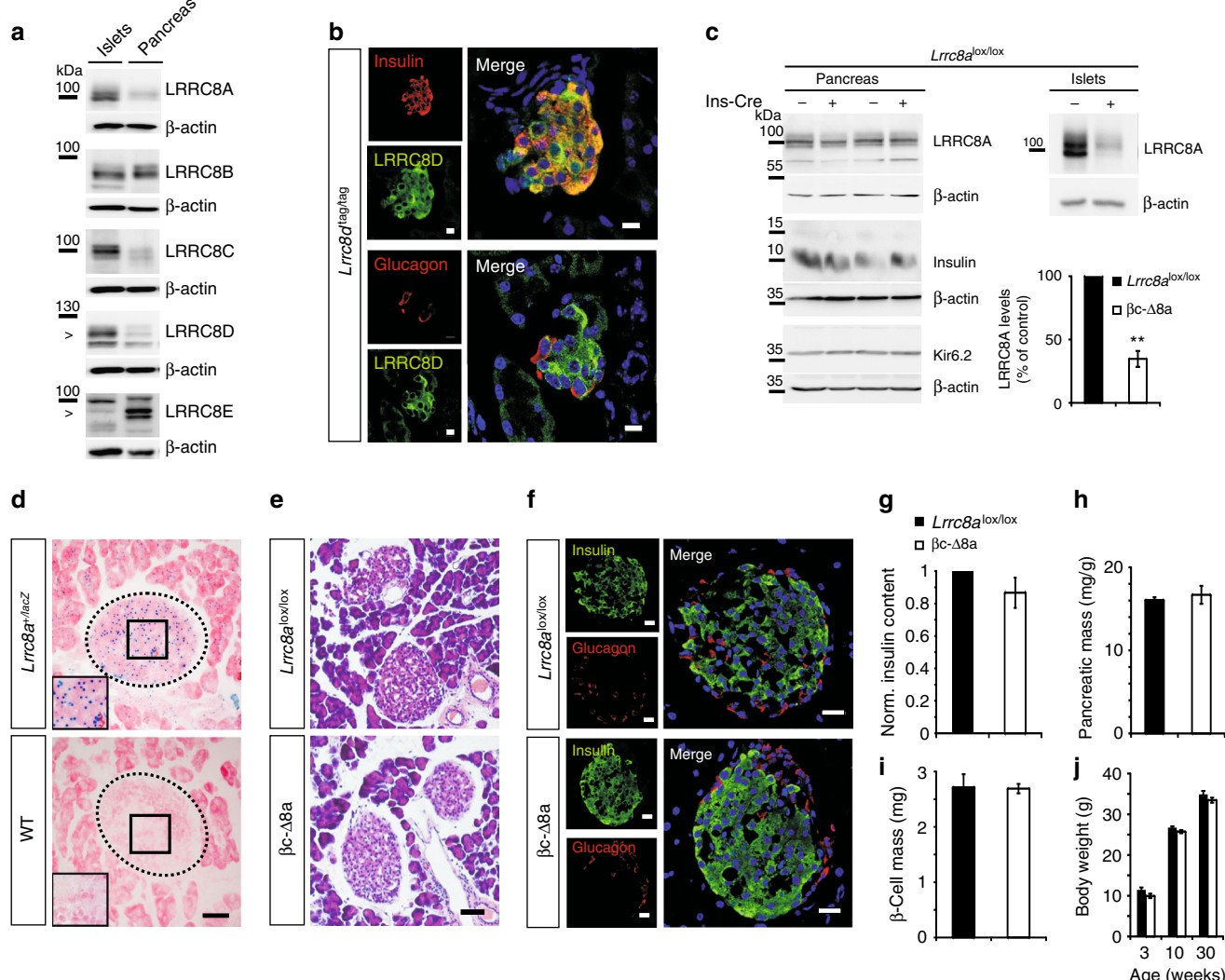

**Fig. 1** LRRC8 proteins in the pancreas and normal islet morphology upon β-cell-specific *Lrrc8a* disruption. **a** Western blot detection of the five VRAC subunits LRRC8A, -B, -C, -D, and −E in lysates of purified islets of Langerhans (left lanes) or total pancreas (right) from wild-type mice. β-actin, loading control. Arrowheads highlight specific bands as determined by previous knock-out controls. **b** Immunofluorescent detection (green) of LRRC8D in pancreatic sections from knock-in mice expressing a LRRC8D-tdTomato fusion protein, co-stained (in red) for insulin (above) or glucagon (below). Left panels, individual channels; right panels, overlays, with co-localization yielding yellow. Note that β-cells express much more LRRC8D than the surrounding tissue. **c** Western blot of lysates from total pancreas or purified islets probed for LRRC8A, insulin, Kir6.2 ($K_{ATP}$ channel subunit) from *Lrrc8a*[lox/lox] mice lacking Cre expression (−) or expressing Cre specifically in β-cells (+). β-actin, loading control. Right bottom, quantification of LRRC8A expression in pancreatic islets normalized to *Lrrc8*[lox/lox] control. $N = 3$ independent experiments. Error bars, mean ± SEM. **$P < 0.01$ (Student's *t*-test). **d** *Lrrc8a* promoter-driven β-gal expression (X-gal staining, blue dots) in islets. Dotted lines highlight islets of Langerhans, insets higher magnification of boxed area. Cells co-stained with eosin Y (pink). **e** Hematoxylin/eosin (H&E) stained formalin-fixed pancreatic sections of control (*Lrrc8a*[lox/lox]) or βc-Δ8a mice (lacking LRRC8A in β-cells). **f** Immunofluorescent staining of pancreatic islets of *Lrrc8a*[lox/lox] and βc-Δ8a animals for insulin (green) and glucagon (red), respectively. **g** Insulin levels of whole pancreas lysates normalized to control (*Lrrc8a*[lox/lox]). **h** Pancreatic mass of *Lrrc8a*[lox/lox] and βc-Δ8a animals determined as ratio of organ to body weight. **i** β-cell mass as defined as ratio of insulin-positive cell area to complete pancreatic tissue area, multiplied by pancreatic weight. **j** Body weight of *Lrrc8a*[lox/lox] (black bars) and βc-Δ8a animals (white bars) at 3, 10 and 30 weeks of age. **g–j** Mean values ± SEM are shown. Differences between genotypes are not significant in unpaired *t*-test: **g** $p = 0.185$, $n = 7$; **h** $p = 0.609$, $n = 3$; **i** $p = 0.968$, $n = 6$. **j** $p = 0.086$ at 3 weeks, $p = 0.05$ at 10 weeks, and $p = 0.214$ at 30 weeks, $n = 16$ (*Lrrc8a*[lox/lox]), $n = 11$ (βc-Δ8a). Scale bars in **b**: 10 μm, in **d–f**: 50 μm

**Hypotonicity and glucose elicit LRRC8/VRAC currents in β-cells.** Exposure of both control (*Lrrc8a*[lox/lox]) and βc-Δ8a β-cells to a hypotonic solution led to similar levels of cell swelling, but β-cells with disrupted *Lrrc8a* lacked the typical slow RVD that was visible in control cells (Fig. 2a). Whole-cell patch-clamp recordings from control β-cells revealed the slow development of outwardly rectifying Cl⁻ currents ($I_{Cl,vol}$) upon exposure to hypotonic medium (Fig. 2b). These currents were almost completely blocked by 20 μM DCPIB, a potent, but non-specific

inhibitor of VRACs[34–37] (Fig. 2b) and were absent from βc-Δ8a β-cells (Fig. 2c–e). As described previously[19,20,23], these currents displayed only weak inactivation at positive potentials (Fig. 2d), a finding we can now attribute to the relatively low expression of the inactivation-promoting subunit LRRC8E[25,28] in islets (Fig. 1a). Two previous reports stated that $I_{Cl,vol}$ of β-cells displays a Cl⁻ > I⁻ selectivity[19,20]. This fueled speculation that the β-cell channel differs from canonical VRAC[38]. Since the strict dependence of $I_{Cl,vol}$ on LRRC8A now identified the channel as

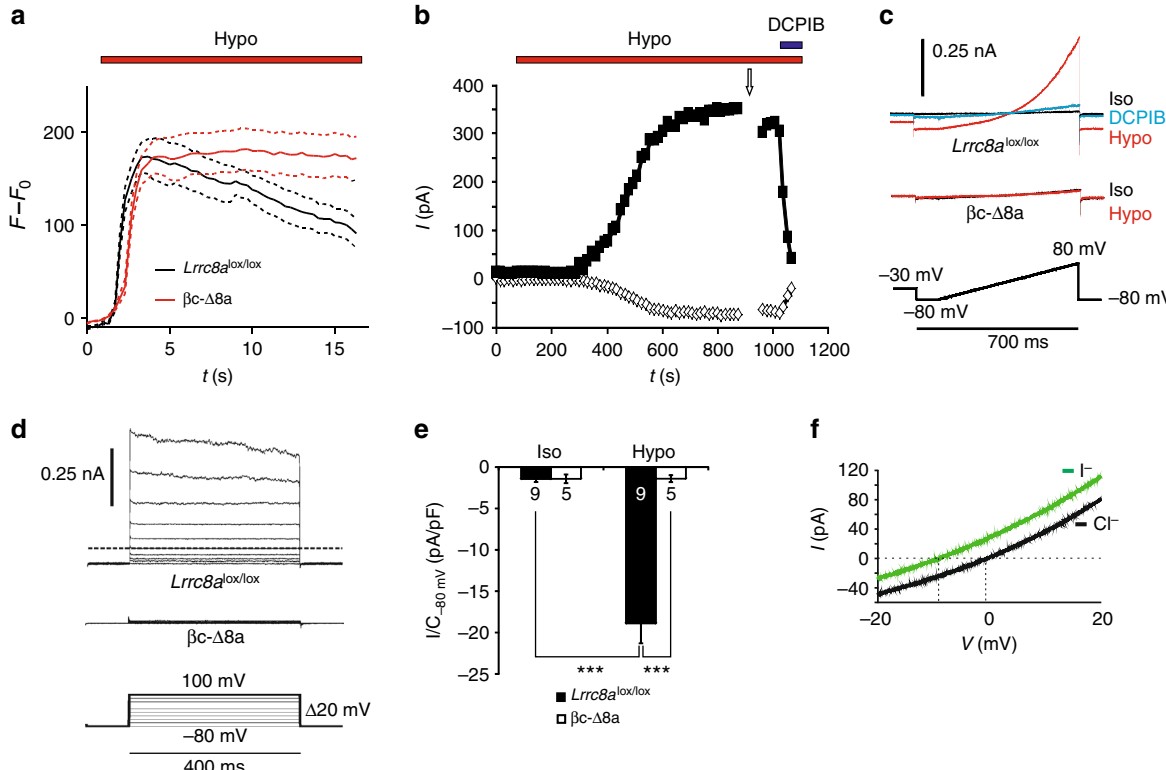

**Fig. 2** Hypotonicity-induced cell swelling and VRAC currents in mouse β-cells. **a** Cell volume as monitored in individual β-cells using calcein fluorescence. The extracellular isotonic solution was replaced by a 30% hypotonic solution (210 mOsm) during the time indicated by the bar. Averaged curves ± SEM (dotted lines), $n = 16$ and $n = 18$ for $Lrrc8a^{lox/lox}$ and βc-Δ8a β-cells, respectively. Data are representative of three independent experiments. **b** Representative time course of $I_{Cl,vol}$ of an isolated primary β-cell provoked by superfusion with hypotonic solution (240 mOsm) without or with 20 μM DCPIB as indicated. Currents elicited at −80 to 80 mV by voltage ramps (see **c**) are plotted. **c** Typical current-voltage (I–V) curves of hypotonicity-provoked $I_{Cl,vol}$ elicited by the ramp protocol shown below, from indicated genotypes and conditions. **d** Representative current traces obtained with the voltage step protocol shown at bottom. The upper traces show control ($Lrrc8^{lox/lox}$) currents after full activation of $I_{Cl,vol}$, taken at the time indicated by arrow in **b**. Even at +100 mV, β-cell $I_{Cl,vol}$ only slightly inactivates. The lower traces are from a βc-Δ8a β-cell exposed for 15 min to hypotonicity. **e** $I_{Cl,vol}$ current densities at −80 mV of $Lrrc8a^{lox/lox}$ and βc-Δ8a β-cells under iso- and hypotonic conditions. Mean values ± SEM; ***$p < 0.0005$ (one-way ANOVA, Tukey's test); number of cells given in bars; iso: isotonic solution (310 mOsm), hypo: hypotonic solution (240 mOsm). **f** Ion selectivity of β-cell $I_{Cl,vol}$. Example traces (from the same cell) obtained with the ramp protocol after maximal $I_{Cl,vol}$ activation, in the presence of extracellular chloride (black trace) or iodide (green trace). The shifts of reversal potentials $E_{rev}$ with $I^-$ to more negative values indicated a permeability ratio $pI^-/pCl^- = 1.44 \pm 0.06$ ($n = 6$)

canonical VRAC (i.e., as LRRC8 channel), we re-investigated this issue (Fig. 2f) and found that β-cell $I_{Cl,vol}$ displays the $I^- > Cl^-$ selectivity that is typical for VRACs[17,25,39].

Exposure of β-cells to high extracellular glucose also activates an outwardly rectifying $Cl^-$ current, which shares several characteristics with $I_{Cl,vol}$[19,20,23]. This activation has been tentatively attributed to β-cell swelling that is observed after exposure to high extracellular glucose[15,40] and that may be caused by osmotic effects of intracellular glucose metabolites[7,40–42]. Indeed, when increasing the glucose concentration of the isotonic superfusate from 3 to 25 mM, cell swelling was observed in both $Lrrc8a^{lox/lox}$ and βc-Δ8a β-cells (Fig. 3a). The extent of swelling was smaller than with large changes of extracellular osmolarity (Fig. 2a) and did not differ significantly between the genotypes. Exposure to 20 mM glucose elicited typical outwardly rectifying, slightly inactivating and DCPIB-sensitive currents in control β-cells (Fig. 3b, c) that were abolished in βc-Δ8a cells (Fig. 3c, d). Hence both hypotonicity- and glucose-induced β-cell currents are mediated by swelling-activated LRRC8/VRAC channels.

**Loss of LRRC8A reduces glucose response of β-cells**. We next investigated the impact of VRAC on β-cell excitation using the perforated patch-clamp technique in the current clamp mode (at

$I = 0$). This allows recordings of the membrane potential with minimal effects on cellular metabolism or ion concentrations. Consistent with VRAC being closed at resting conditions[17,39] (Fig. 2b), the resting membrane potential did not differ between $Lrrc8a^{lox/lox}$ and βc-Δ8a β-cells, neither at 1 nor at 5 mM external glucose (Fig. 4a). When exposed to external hypotonicity (220 mOsm), β-cells from $Lrrc8a^{lox/lox}$, but not from βc-Δ8a mice, slowly depolarized (Fig. 4b–d), presumably by an efflux of $Cl^-$ through LRRC8 channels, which activate under these conditions with a similarly slow time course (Fig. 2b). This depolarization was more pronounced at 5 mM (Fig. 4b, c) than at 1 mM glucose (Fig. 4e, f). This difference can be explained by a partial inhibition of $K_{ATP}$ at 5 mM glucose. Although not sufficient to significantly depolarize β-cells when VRAC is closed or absent (in βc-Δ8a cells) (Fig. 4a), this inhibition reduces the hyperpolarizing $K^+$ current that opposes the depolarizing $Cl^-$ current through VRAC. In a sizeable fraction of $Lrrc8a^{lox/lox}$, but not βc-Δ8a β-cells, hypotonicity elicited membrane potential spiking (Fig. 4g, h).

Increasing external glucose concentration from 1 to 15 mM induced action potentials in both $Lrrc8a^{lox/lox}$ and βc-Δ8a β-cells (Fig. 5a, b). However, cells devoid of VRAC needed on average ~100 s longer to reach the spiking threshold, although this difference was not statistically significant with our sample size

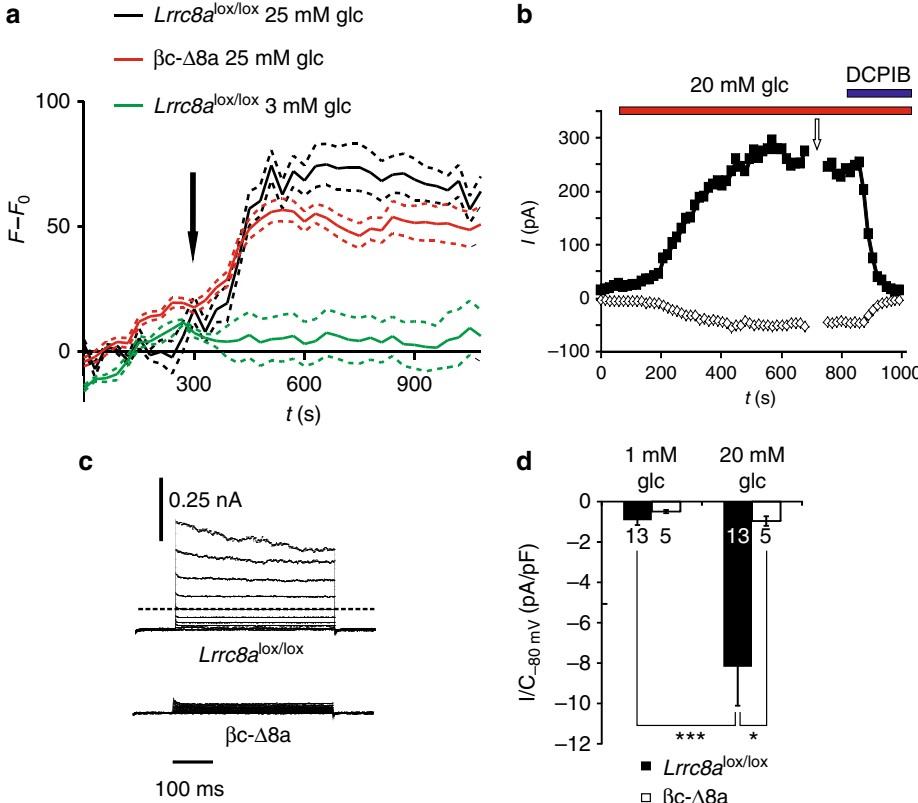

**Fig. 3** Cell swelling and induction of LRRC8/VRAC currents in β-cells by high extracellular glucose. **a** Cell volume as monitored in individual β-cells loaded with calcein. Cell bath was switched from a 3 mM to a 25 mM glucose-containing isotonic solution at the time indicated by the arrow. Shown are mean values ± SEM (dotted lines), $n = 15$ and $n = 23$ for $Lrrc8a^{lox/lox}$ and βc-Δ8a β-cells, respectively, and are representative of three independent experiments. Green trace corresponds to bath change without changing the glucose concentration, a control to exclude perfusion artefacts ($n = 6$). **b** Time course of anion current activation by superfusion with 20 mM glucose in isotonic saline (310 mOsm). Application of 20 μM DCPIB blocked the currents. Minimal and maximal currents elicited at −80 or 80 mV by voltage ramps (as in Fig. 2c) are plotted. **c** Representative current traces obtained after >10 min superfusion with 20 mM glucose from the indicated genotypes. Currents were elicited by the voltage protocol shown in Fig. 2d. The upper traces show currents at the time indicated by the arrow in **b**. **d** $I_{Cl,vol}$ current densities at −80 mV of $Lrrc8a^{lox/lox}$ and βc-Δ8a β-cells under indicated conditions. Mean currents ± SEM; *$p < 0.05$; ***$p < 0.0005$ (one-way ANOVA, Tukey's test); number of cells indicated in bars

(Fig. 5c). The frequency and amplitude of glucose-induced action potentials were unchanged (Fig. 5d). β-cells began to spike before they depolarized to the degree observed with hypotonicity (Fig. 4), presumably because the increased membrane resistance caused by partial $K_{ATP}$ closure renders the positive feedback between voltage-dependent $Na^+$- and $Ca^{2+}$-channel opening and depolarization more efficient. In contrast to glucose, which requires cellular uptake and metabolism to exert its inhibitory and stimulatory effects on $K_{ATP}$ and VRAC, respectively, the $K_{ATP}$ inhibitor tolbutamide almost instantly elicited β-cell spiking that was indistinguishable between the genotypes (Fig. 5a, b, e). This agrees with unchanged Kir6.2 protein expression (Fig. 1c) and suggests that the activity of this pivotal channel is not affected by the loss of VRAC. As another read-out for β-cell stimulation, we determined the rise of intracellular $Ca^{2+}$ ($[Ca^{2+}]_i$), the final trigger for insulin exocytosis. Mirroring the effect on β-cell excitability, 15 mM glucose increased $[Ca^{2+}]_i$ in both genotypes, but with a significantly longer delay in βc-Δ8a cells (Fig. 5f–h). A tendency for a delayed glucose response of $Lrrc8a^{-/-}$ β-cells became already apparent in the 6–8 mM glucose concentration range (Supplementary Fig. 3). The peak $Ca^{2+}$-response, however, did not differ significantly between the genotypes (Fig. 5i). To investigate whether VRAC influences β-cell excitability beyond the initial 15 min investigated so far, we studied the effect of 10 mM glucose on $Ca^{2+}$ oscillations of intact islets that were

preincubated for 30 min with this glucose concentration before starting the measurements. Islets from both genotypes showed the typical range of $Ca^{2+}$ oscillation patterns[43] (Supplementary Fig. 4), with no obvious difference between the genotypes. To conclude, although VRAC is not required for glucose to excite β-cells, it increases their glucose sensitivity during the early phase of glucose stimulation.

**$Lrrc8a$ disruption reduces insulin secretion**. We next asked whether VRAC modulates insulin secretion. Supernatants from single islets from $Lrrc8a^{lox/lox}$ and βc-Δ8a mice were collected 30 min after stimulation and their insulin content determined by ELISA (Fig. 6a). There was no difference in insulin release between the genotypes in the presence of 3.3 mM glucose. Depolarization of β-cells by increasing $[K^+]_o$ from 5 to 45 mM, or by exposure to 300 μM tolbutamide, increased insulin secretion about 15- and 5-fold, respectively, irrespective of the $Lrrc8a$ genotype. Increasing glucose concentration to 25 mM enhanced insulin release about eightfold and sixfold with $Lrrc8a^{lox/lox}$ and βc-Δ8a islets, respectively. However, this apparent difference failed to reach statistical significance ($p = 0.11$). A similar degree of stimulation and borderline difference between the genotypes ($p = 0.08$) was observed when islets were exposed to both 25 mM glucose and 300 μM tolbutamide. The absence of a marked effect

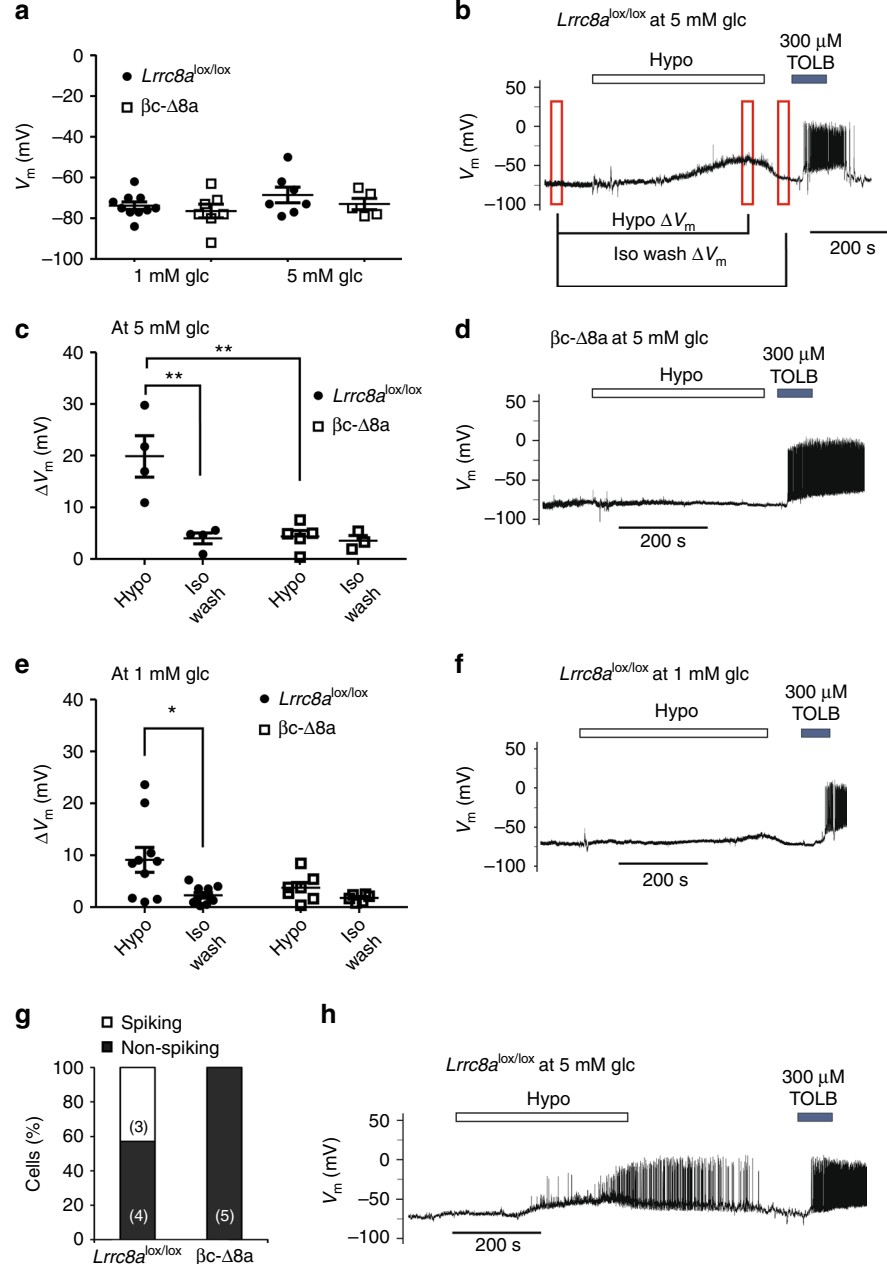

**Fig. 4** Hypotonicity-induced depolarization of β-cells. Gramicidin-perforated patch-clamp recordings of primary mouse β-cells. **a** Resting membrane potential of control $Lrrc8a^{lox/lox}$ or βc-Δ8a β-cells at 1 or 5 mM glucose concentrations. Mean values ± SEM indicated by error bars. **b** Representative trace for hypotonicity-induced depolarization of the membrane potential $V_m$ in control $Lrrc8a^{lox/lox}$ animals at 5 mM glucose. TOLB, tolbutamide. **c** $\Delta V_m$ after applying hypotonicity, or following washout with isotonic solution of $Lrrc8a^{lox/lox}$ or βc-Δ8a cells at 5 mM glucose, calculated from data as indicated in **b**. Spiking cells (see **g** and **h**) were excluded from analysis. **d** Representative voltage trace showing lack of hypotonicity-induced depolarization of βc-Δ8a β-cells at 5 mM glucose. **e** $\Delta V_m$ of $Lrrc8a^{lox/lox}$ or βc-Δ8a cells as in **c**, but at 1 mM glucose. **f** Representative trace for $Lrrc8a^{lox/lox}$ β-cells challenged with hypotonicity in the presence of 1 mM glucose. **g** Percentage of cells responding with electrical activation following hypotonic stimulation. **h** Voltage trace of an $Lrrc8a^{lox/lox}$ β-cell responding with electrical activity to hypotonicity in the presence of 5 mM glucose. $*p < 0.05$, $**p < 0.01$ (one-way ANOVA, Tukey's test)

of VRAC ablation on insulin secretion agrees with the limited effect on glucose-induced excitation and $Ca^{2+}$-transients of β-cells, which was apparent only in a 5–10 min time window (Fig. 5c, f–h, Supplementary Fig. 3). Indeed, when measured only within the first 8 min of adding 25 mM glucose, $Lrrc8a^{lox/lox}$ islets secreted about twice as much insulin than their βc-Δ8a counterparts ($p = 0.008$) (Supplementary Fig. 5). However, these high glucose concentrations are unlikely to be reached in vivo. We therefore examined whether VRAC modulates insulin secretion

with more physiological glucose stimuli, which was suggested by the effect of 15 mM (Fig. 5a–c, f–h) and possibly even lower (Supplementary Fig. 3) glucose concentrations on β-cell excitability. Indeed, a significant, roughly 50% decrease of insulin secretion by βc-Δ8a islets in an 8 min time window was observed also with 10 mM glucose (Fig. 6b). There was no difference in the response to tolbutamide or high potassium under identical conditions, confirming that canonical $K_{ATP}$ signaling is not affected (Fig. 6b). We finally tested whether the deletion of VRAC in β-

cells affects the regulation of blood glucose in vivo. Whereas serum concentrations of glucose were not different between the genotypes ($Lrrc8a^{lox/lox}$, $163 \pm 12$ mg/dl; βc-Δ8a, $162 \pm 9$ mg/dl; $n = 11$), βc-Δ8a mice were abnormal in glucose tolerance assays (Fig. 6c). Mice of both genotypes were injected intraperitoneally with glucose solutions and their blood glucose levels were followed over 2 h. Compared to $Lrrc8a^{lox/lox}$ control mice, βc-Δ8a mice displayed significantly higher serum glucose concentrations

30 and 60 min after the glucose load. By contrast, mice of either genotype behaved similarly in insulin tolerance tests (Fig. 6d), demonstrating that β-cell-specific disruption of VRAC, as expected, does not influence peripheral glucose uptake. Importantly, $Lrrc8a^{+/+}$; Ins-Cre[+] and $Lrrc8a^{lox/lox}$; Ins-Cre[−] control groups behaved indistinguishably in these assays, showing that a reported interference of the Ins2-Cre construct itself[44] is not of concern here. Finally, glucose-stimulated insulin secretion in vivo

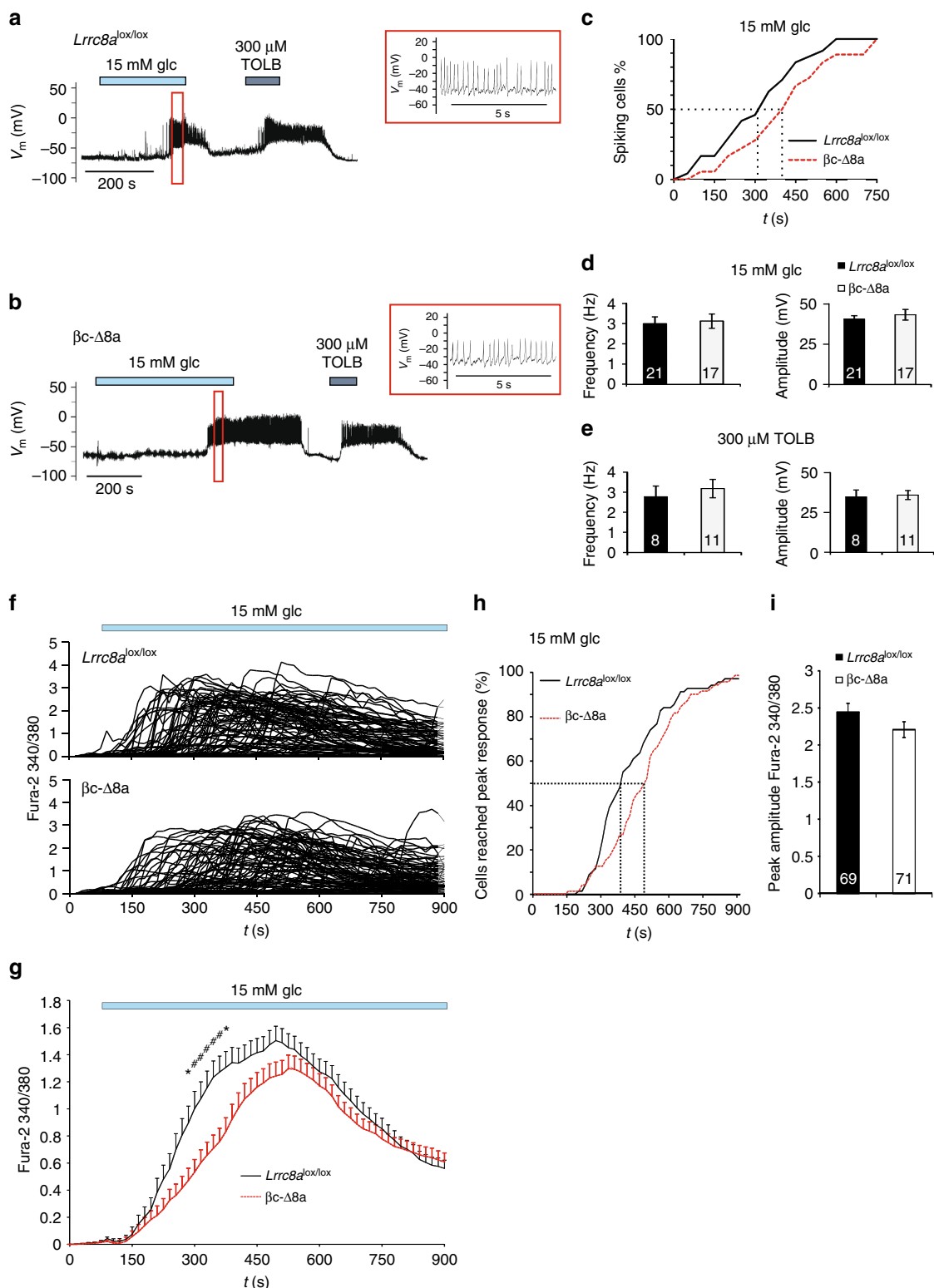

**Fig. 5** Glucose-induced electrical activity and intracellular $Ca^{2+}$ response in *Lrrc8a*[lox/lox] and βc-Δ8a cells. **a, b** Representative voltage traces for glucose- and tolbutamide-induced electrical activity of control *Lrrc8a*[lox/lox] (**a**) and βc-Δ8a (**b**) β-cells obtained with gramicidin-perforated patches in the current clamp mode. Enlarged traces are shown in red boxes, with corresponding time periods indicated by boxes in the complete trace at left. **c** Percentage of electrically active cells plotted against time after glucose application. The response of βc-Δ8a cells seemed delayed by about 100 s (dotted lines), but the difference between genotypes failed to be statistically significant with 24 control and 18 βc-Δ8a β-cells. **d** Mean frequency and amplitude of spikes during bursts of action potentials elicited by 15 mM glucose. **e** Mean frequency and amplitude of spikes elicited by 300 µM tolbutamide. **f** Individual traces for Fura-2 fluorescence ratios elicited by excitation at $\lambda = 340$ and 380 nm (indicative of $[Ca^{2+}]_i$). 15 mM glucose was added at $t = 60$ s. **g** Mean ratio of Fura-2 fluorescence over time. Time points at which differences between the two genotypes reached statistical significance are indicated. *$p < 0.05$, #$p < 0.01$ (two-way ANOVA, Bonferroni multiple comparisons); mean values ± SEM, 69 control and 71 βc-Δ8a β-cells. **h** Cumulative plot of cells that have reached peak levels of $[Ca^{2+}]_i$ after addition of 15 mM glucose as function of time. The difference is statistically significant ($p = 0.005$, Kolmogorov-Smirnov test). **i** Peak value of calcium response of β-cells of both genotypes. The difference is not significant. Number of cells indicated in columns

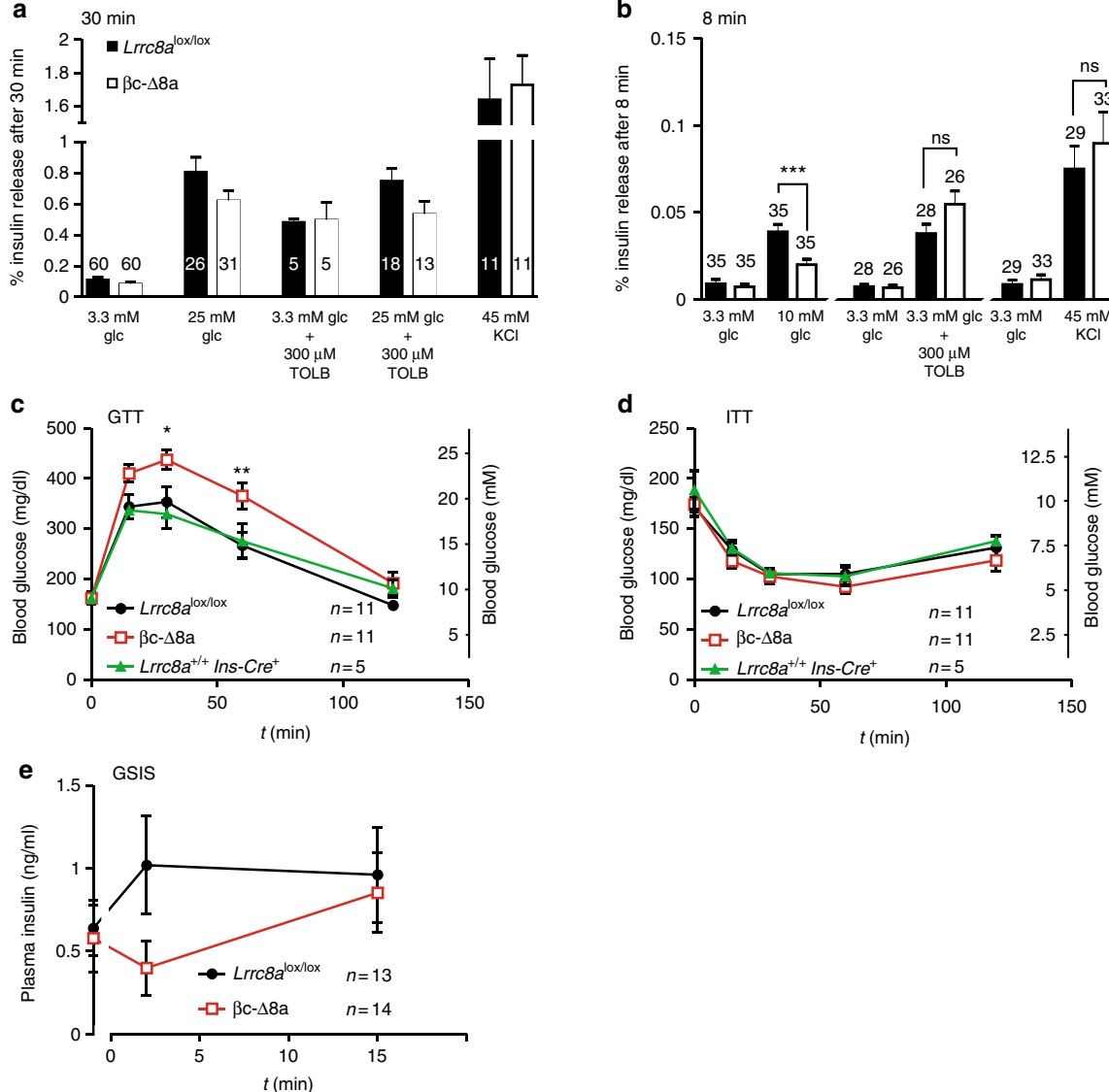

**Fig. 6** Effect of LRRC8A/VRAC disruption on insulin secretion in vitro and in vivo and on glucose tolerance. **a** Insulin release from single islets stimulated by high glucose, tolbutamide, and potassium. Islets were incubated for 30 min under the indicated conditions and insulin concentrations in the supernatant were measured. Number of islets is indicated in bars (derived from 5 mice per genotype (25 mM glc), 1 mouse per genotype (TOLB), 3 mice per genotype (KCl), 2 mice per genotype (TOLB + glc)). **b** Insulin release of single islets under the indicated conditions during the first 8 min of incubation. Number of islets (from three mice per genotype and condition) is indicated in bars. ***$p < 0.001$ (one-way ANOVA, Tukey's test). **c** Glucose tolerance test. Changes in blood glucose levels following intraperitoneal injection of glucose (2 mg/g body weight) in *Lrrc8a*[lox/lox], *Lrrc8a*[+/+];Ins-Cre[+] and βc-Δ8a animals was followed over time. **d** Insulin tolerance test. Changes in blood glucose levels over time following intraperitoneal injection of insulin (0.75 U/kg body weight). **c, d** Number of mice (male adults, 10–15 weeks old) for each experiment is indicated. *$p < 0.05$, **$p < 0.01$ (two-way ANOVA, Bonferroni multiple comparisons); mean values ± SEM. **e** Glucose-stimulated insulin secretion in vivo. Changes in the plasma insulin concentration after intraperitoneal injection of a glucose load (2 mg/g body weight) over time. Number of mice (adult males and females, 10–15 weeks of age) is indicated. Unpaired $t$-test for insulin levels at 2 min after glucose injection yields $p = 0.07$

(Fig. 6e) suggested that early insulin secretion of βc-Δ8a mice is impaired, agreeing with our results on β-cell excitability and insulin secretion in vitro. Although, like in every gene disruption experiment, we cannot totally exclude secondary effects, we conclude that LRRC8/VRAC channels of β-cells modulate insulin secretion and the regulation of blood glucose.

## Discussion

Serum glucose concentration is subject to complex regulatory mechanisms mainly involving glucose-mobilizing glucagon and glucose-lowering insulin. These hormones are produced and secreted by pancreatic islet α- and β-cells, respectively. Their secretion is not only controlled by serum glucose, but also by other hormones such as incretins, neurotransmitters, and paracrine/autocrine mechanisms within the islet[1,45–47]. Glucose sensing by β-cells crucially involves inhibition of $K_{ATP}$ channels by intracellular ATP/ADP ratios, which rise owing to increased glucose uptake and metabolism[2]. Our work reveals an important modulatory role of LRRC8/VRAC channels in β-cell stimulus-secretion coupling. These chloride channels depolarize β-cells in response to glucose-induced β-cell swelling, thereby enhancing the glucose-induced increase in cytoplasmic $Ca^{2+}$ and insulin secretion. The presence of neurotransmitter-permeable LRRC8D-containing VRACs in β-cells raises the possibility that they may also be involved in autocrine and paracrine signaling in islets.

Evidence has accumulated over many years that ion channels other than $K_{ATP}$ may be involved in glucose-induced β-cell depolarization and insulin secretion[3]. Prime candidates are $Cl^-$ channels since their opening leads to a depolarizing $Cl^-$ efflux in the presence of the high intracellular $Cl^-$ concentration of β-cells[11,13]. Recently the cystic fibrosis transmembrane conductance regulator CFTR has been implicated in modulating the glucose response and insulin secretion of β-cells[9]. Together with the destruction of pancreatic tissue in certain forms of the disease, reduced CFTR-stimulated insulin secretion might contribute to diabetes in some patients with cystic fibrosis. CFTR needs intracellular ATP for gating, but it is unclear whether glucose-induced increase in $[ATP]_i$ can account for the apparently glucose-induced opening of this cAMP-gated channel. VRAC has emerged as another attractive candidate for a $Cl^-$ channel that modulates insulin secretion[7]. However, the anion selectivity previously reported for $I_{Cl,vol}$ in β-cells[19,20] does not fit to canonical VRAC/LRRC8 channels[17,25,39,48] and the mystery surrounding its molecular identity[21,49] for a long time has precluded conclusive studies both in vitro and in vivo.

We have now demonstrated conclusively that $I_{Cl,vol}$ of β-cells is mediated by LRRC8 channels, i.e., canonical VRACs displaying a $I^- > Cl^-$ selectivity. Opening LRRC8 channels by hypotonicity can depolarize β-cells up to the threshold for action potential firing. Although the extent of hypotonicity used in our experiments is clearly non-physiological, these experiments provide proof of principle that swelling-activation of LRRC8 channels can depolarize β-cells to voltages that depend on the magnitude of glucose-induced $K_{ATP}$ inhibition. The situation is more complex with exposure to high glucose. It both inhibits $K_{ATP}$ channels by the slow increase in ATP/ADP ratios, and slowly activates LRRC8 channels, probably by cell swelling that may be due to osmotic effects of several glucose metabolites including lactate[7,40,42]. Glucose activation of VRAC and inhibition of $K_{ATP}$ have a similar time course that is largely determined by the accumulation of glucose metabolites. As shown by the delayed, but otherwise normal response of βc-Δ8a cells to high glucose, VRAC is clearly not required for glucose-induced excitation of β-cells, but modulates their glucose sensitivity. Accordingly a marked effect of VRAC on insulin secretion was only observed at 8 min of glucose addition to isolated islets. When sampled over 30 min, differences in insulin secretion failed to reach significance levels. Importantly, however, β-cell-specific disruption of VRAC led to significantly impaired glucose tolerance and glucose-induced insulin secretion appeared to be decreased in vivo.

While this manuscript was under review, Sah and coworkers asserted[50] that the LRRC8A subunit (renamed "SWELL1") is a "glucose sensor" required for glucose-induced excitation and insulin secretion of β-cells. Both studies coincide in that VRAC plays a role in glucose sensitivity of β-cells in vitro and in vivo and agree in several other aspects. However, in stark contrast to our study, Kang et al. reported that glucose-induced $Ca^{2+}$ transients and insulin secretion were abolished in $Lrrc8a^{-/-}$ β-cells, rather than only being delayed or decreased as found by us. The suggested absolute requirement for VRAC[50] is surprising in view of the rather mild phenotypes of β-cell-specific $Lrrc8a^{-/-}$ mice and unchanged resting glucose levels in either study, and also because the major, canonical $K_{ATP}$ pathway for glucose sensing remains intact upon $Lrrc8a$ disruption. Moreover, consistent with VRAC being closed at rest, both studies reported that the resting potential of $Lrrc8a^{-/-}$ β-cells is unchanged. This virtually eliminates the possibility that β-cell hyperpolarization counteracts the depolarizing effect of $K_{ATP}$ closure in $Lrrc8a^{-/-}$ mice. A possible explanation for the discrepancy between both studies might be found in methodological differences. Whereas we used β-cells from βc-Δ8a mice, Kang et al.[50] more acutely disrupted $Lrrc8a$ with adenoviral transduction of Cre-recombinase into $Lrrc8a^{lox/lox}$ cells, or alternatively reduced $Lrrc8a$ expression by transduction of shRNA. Whereas in our study the lack of VRAC might have been compensated by altered expression of other channels (although the $K_{ATP}$ pathway appeared unchanged), the acute viral overexpression of either Cre-recombinase or shRNA by Kang et al.[50] might have caused secondary changes that further decreased the glucose sensitivity of $Lrrc8a^{-/-}$ β-cells. Another decisive factor may be the rather short exposure of β-cells to glucose used by Kang et al.[50], who thus may have missed the delayed glucose response of KO cells found here.

Several aspects of β-cell LRRC8/VRAC channels need to be considered. First, VRAC-dependent cell volume regulation is not essential for the survival, development, and overall function of β-cells, probably because cells dispose of several redundant mechanisms for regulating their volume[16]. Importantly, β-cell-specific $Lrrc8a$ disruption did not affect pancreas and islet morphology, β-cell mass, insulin content, Kir6.2 expression, the response to tolbutamide, and did not cause inflammation. Hence our results are unlikely to be influenced by developmental or compensatory changes. Second, although VRAC needs basal levels of ATP for channel activity[17,51], it is not activated by intracellular ATP. Glucose activation of VRAC is most likely caused by osmotic cell swelling due to glucose metabolites[7,40,42], a notion that is bolstered by our study. Multiple mechanisms have been proposed to explain volume-activation of VRAC[17,49], including a direct activation by low intracellular ionic strength[27,52,53]. Third, VRAC might only have a transient role in glucose-stimulated insulin secretion because its opening will lead to an efflux of chloride and organic osmolytes such as taurine[25,54], resulting in RVD and a subsequent downregulation of VRAC activity. Although we could not detect RVD in glucose-swollen β-cells over a period of 12 min (Fig. 3a), which might be owed to continued generation of intracellular glucose-derived osmolytes, a transient role of VRAC in the initial response to glucose is indeed suggested by our experiments (Fig. 5c, g, h, Fig. 6a, b). Fourth, VRAC may play a role in autocrine and paracrine signaling in pancreatic islets. LRRC8 channels not only conduct chloride, but also organic compounds including osmolytes, drugs, and neurotransmitters[25,29,30,55]. Transport of these

compounds is determined by the LRRC8 subunit composition[29,30]. Intriguingly, LRRC8D, a subunit important for the transport of all organic compounds tested so far[29,30], shows much stronger expression in islets than in whole pancreas (Fig. 1a) and is abundantly expressed in β-cells (Fig. 1b). LRRC8D-containing VRACs conduct, for instance, taurine[29], an agonist of glycine and GABA receptors[56], GABA[30], glycine[57] and glutamate[30]. β-cells express GABA-synthesizing glutamate decarboxylases[58] and display sizeable cytoplasmic concentrations of GABA[58–61]. Therefore, in addition to vesicular discharge[12,46,62], neurotransmitters and regulators might be released from β-cells through LRRC8/VRAC channels. Both α- and β-cells express glycine and GABA receptors[12,61–63] that may influence glucose homeostasis by stimulating β-cells in an autocrine fashion or by exerting a paracrine effect on glucagon secretion by α-cells. In addition to the direct depolarization by VRAC-mediated $Cl^-$ efflux, VRAC might additionally influence systemic glucose homeostasis by autocrine/paracrine effects within islets.

## Methods

**Animals.** Animals were housed under standard conditions in the MDC animal facility according to institutional guidelines. All animal experiments were approved by German authorities (LAGeSo) in a letter to the MDC. Mice heterozygous for the targeted Lrrc8a allele (Lrrc8a[tm2a(EUCOMM)Hmgu]) were generated by injecting targeted ES cells (obtained from EUCOMM (European Conditional Mouse Mutagenesis Program) consortium) into blastocysts (MDC Transgenic Facility). Lrrc8a[tm2a(EUCOMM)Hmgu] animals were crossed to Cre- or Flp-recombinase expressing deleter mice[64,65] to generate two different Lrrc8a mouse models (Supplementary Fig. 1). For β-cell-specific Lrrc8a deletion, Lrrc8a[lox/lox] mice were crossed to Ins2-Cre (Tg(Ins2-cre)23Herr) mice, which express the Cre-recombinase under the control of the rat insulin2 promotor[31]. Ins2-Cre animals were also crossed to Rosa26-Cre reporter line (B6.Cg-Gt(ROSA)26Sor[tm9(CAG-tdTomato)Hze]/J)[66] to verify cell-type specificity of Cre expression (Supplementary Fig. 6a). Although the Ins2-Cre (Tg(Ins2-cre)23Herr) mouse expresses the recombinase also in sparse cells scattered over the brain[67,68], which we confirmed by crossing this strain with reporter mice (Supplementary Fig. 6b), this is not a concern since Kang et al.[50] used the recommended Ins1Cre[ERT2] strain to delete Lrrc8a and obtained results in vivo similar to ours. Knock-in mice expressing an LRRC8D-tdTomato fusion protein from the endogenous Lrrc8d promoter were generated by CRISPR-Cas9 mediated recombination in fertilized mouse oocytes that were implanted into foster mothers by the MDC Transgenic Facility. Sequence encoding the tdTomato fluorescent protein[69] was added to the carboxy-terminus of LRRC8D. Control experiments indicated that the tag did not change the electrophysiological properties or expression of LRRC8D in transfected cells. The generation and characterization of these mice will be described in detail elsewhere. The fluorescence of the fusion protein was insufficient for detection in tissue sections, requiring immunohistochemistry with anti-RFP-antibodies visualization of the fusion protein. All mice were in a C57/BL6 genetic background.

**Antibodies and western blots.** Rabbit polyclonal antibodies against individual LRRC8 subunits had been generated in our laboratory and their specificity had been confirmed in western blots using KO cell lines as controls as described[25,29,30]. The following commercial antibodies were used: guinea pig anti-insulin (Dako, A0564; 1:500), mouse anti-glucagon (Abcam, ab10988; 1:1000), mouse anti-β-actin (Sigma, A2228; 1:2000), rabbit anti-RFP (Rockland, 600401379; 1:1000), rabbit anti-Kir6.2 (Abgent, AG1143; 1:200), goat anti-NLRP3 (Lifespan Biosciences, LS-B1766; 1:200), rabbit anti-active caspase 3 (Promega; G748A1; 1:400). Secondary antibodies were from Molecular Probes (coupled to Alexa-488 or Alexa-555 or Alexa 633), or from Jackson ImmunoResearch (coupled to horseradish peroxidase). Quantification of western blots used ImageJ. LRRC8A levels were normalized to actin controls on the same blot. Uncropped western blots are shown in Supplementary Fig. 7.

**Histology and immunohistochemistry.** For histology, deeply anesthetized mice were perfused with PBS, followed by PBS containing 4% paraformaldehyde (PFA). Dissected pancreas were postfixed overnight in 4 % PFA, dehydrated in a graded isopropanol series, and then incubated in isopropanol/paraffin (1:1) mixture overnight at 65 °C. The tissue was embedded in paraffin and cut in 8 μm sections which were stained with hematoxylin and eosin Y (H&E) according to standard procedure. To prepare frozen tissue sections, tissue from perfused animals was postfixed for 45 min in 4% PFA, incubated in 30% sucrose overnight at 4 °C, and embedded in Tissue-Tek O.C.T. (Sakura). Frozen sections of 8 μm were cut using a cryostat (Microm HM 560, Thermo Fisher Scientific). For X-Gal staining, sections were first stained with eosin Y, washed in 0.1 M phosphate buffer (pH 7.4, containing 2 mM $MgCl_2$, 5 mM EGTA, 0.02% NP-40, 0.01% Na-deoxycholate), and

then incubated for 15 h at 37 °C in 0.1 M phosphate buffer (pH 7.4) containing X-Gal 1 mg/ml, 2 mM $MgCl_2$, 5 mM EGTA, 0.2% NP-40, 0.1% Na-deoxycholate, 5 mM $K_3Fe(CN)_6$, 5 mM $K_4Fe(CN)_6$. Images of H&E or X-Gal stainings were taken with an AxioCam MRc 5 camera on an Axiophot microscope using the ZEN software (all from Zeiss). Immunohistochemistry was performed on frozen tissue sections permeabilized with 0.2% Triton X-100 in PBS, blocked for 30 min in blocking buffer (PBS containing 0.1% Triton X-100, 3% BSA) and incubated overnight at 4 °C with primary antibodies in blocking buffer. Sections were incubated with secondary antibodies and counterstained with DAPI (Invitrogen) for 45 min at room temperature. Confocal images were taken with a Zeiss LSM 510 microscope using the ZEN software.

**Islet purification and primary cell culture.** Eight- to 18-week-old mice of either sex were killed by cervical dislocation. The bile duct was clamped before inflating the pancreas with HBSS. The inflated pancreas was dissected, cut into smaller pieces and digested for 11 min at 37 °C using 15 ml HBSS containing 20 mg collagenase from *Clostridium histolyticum* (Sigma, C9407–5G) with vigorous shaking every 2 min. Digestion was stopped with ice-cold HBSS and islets were further purified from exocrine tissue using a Histopaque gradient and centrifugation for 10 min at 800×g (Heraeus Multifuge 3 l). Histopaque 1119 (Sigma, 11191) was diluted with HBSS to obtain solutions with different densities (1.119, 1.1, 1.108, 1.106 g/ml). Islets were transferred to RPMI 1640 and manually isolated under microscopic control. Purified islets were incubated in RPMI 1640 (containing 10% FCS, 1% pen/strep) at 37 °C and 5% $CO_2$ overnight before performing further experiments. Primary β-cell cultures were prepared by dispersing the purified islets using an enzyme-free cell dissociation solution (Millipore, S-004-B) and cultured in RPMI 1640 (10% FCS, 1% pen/strep). Single isolated β-cells were used for patch-clamp and calcium measurement.

**Insulin secretion in vitro.** Single islets of similar size were transferred to a 96-well plate and were subsequently incubated at 37 °C on a shaking platform (300 r.p.m.) for 30 min each in solutions containing different stimuli. Afterwards islets were lysed and released insulin was normalized to the total islet insulin content. Standard secretion buffer contained (in mM) 98 NaCl, 0.9 $CaCl_2$, 2.7 KCl, 1.5 $KH_2PO_4$, 0.5 $MgCl_2$, 8 $Na_2HPO_4$, 20 HEPES, 0.2% BSA, pH 7.4 with NaOH. Different concentrations of glucose (3.3, 10 or 25 mM) or 300 μM tolbutamide were added and the osmolarity of the solutions was balanced by adding mannitol (290 mOsm). Islets that showed morphological changes indicative of lysis or cell death at the end of the secretion period were excluded from the analysis. Islets were lysed afterwards to normalize secreted insulin amounts to the total insulin content. Insulin levels in the supernatant or from total pancreatic lysates were measured using an ultra-sensitive insulin ELISA (Alpco, 80-INSMSU-E01) according to manufacturer's instructions. The genotype of islets was revealed only after the end of the experiment.

**Cytoplasmic $Ca^{2+}$ measurements.** Single β-cells or intact islets were seeded on poly-L-lysine (PLL) coated glass bottom live-cell dishes 24 h before. Cells or islets were washed with bath solution containing (in mM) 120 NaCl, 5 KCl, 1 $MgCl_2$, 2.5 $CaCl_2$, 20 HEPES, and 3 glucose (pH 7.4). Bath solution containing 6, 8, 10, or 15 mM glucose were used for stimulation and osmolarity of the solutions was compensated by mannitol (290 mOsm). Cells or islets were loaded with Fura-2 AM (Invitrogen) (5 μM) at 37 °C for 30 min. After washing with bath solution, live-cell dishes were mounted on an Axiovert 200 microscope (Zeiss) and alternatively excited at $\lambda$ = 340 or 380 nm (Polychrome V TILL Photonics) while emission signals were recorded at $\lambda$ = 510 nm. The ratio of 340/380 fluorescence was calculated and values were normalized to $F_0$. Solutions containing 6, 8, or 15 mM glucose were added after 60 s and the fluorescence was measured every 15 s. To measure $Ca^{2+}$ oscillations in intact islets, these were already stimulated with 10 mM glucose solution 30 min before the experiment. $Ca^{2+}$ oscillations were measured in the continued presence of 10 mM glucose. Fura-2 fluorescence ratios were measured every 3 s for 20 min.

**Electrophysiology.** Single β-cells, being bigger than other pancreatic cells, were identified by their size and granular appearance and showed a membrane capacitance of >5 pF. They were cultured on PLL-coated coverslips 24–48 h before recording. Whole-cell patch-clamp recordings were performed using isotonic solution containing (in mM) 130 NaCl, 5 KCl, 1 $MgCl_2$, 2.5 $CaCl_2$, 20 HEPES, 20 mannitol, pH 7.4 with NaOH (310 mOsm) or hypotonic solution containing (in mM) 105 NaCl, 5 CsCl, 1 $MgCl_2$, 2.5 $CaCl_2$, 20 HEPES, pH 7.4 (240 mOsm). To determine ion selectivity, 105 mM NaCl was substituted by 105 mM NaI. Glucose-activated $I_{Cl,vol}$ was measured in isotonic solution with 20 mM glucose substituting mannitol (pH 7.4, 310 mOsm). The patch pipette solution contained (in mM) 100 CsCl, 20 tetraethylammonium-Cl, 10 HEPES, 4 MgATP, 5 EGTA, 1.93 $CaCl_2$ (calculated free $Ca^{2+}$: 50–100 nM), 25 mannitol (pH 7.2 with CsOH, 290 mOsm). The holding potential was set to −30 mV. A ramp protocol from −80 to 80 mV (Fig. 2b) was run every 15 s to record the time course of $I_{Cl,vol}$ activation. Following complete activation of $I_{Cl,vol}$, a step voltage protocol was applied (−80 to 100 mV in 20 mV steps every 5 s (Fig. 2c)). Relative anion permeability was calculated as described[25].

Membrane potentials were measured by gramicidin-perforated patch-clamp recordings in the current clamp mode ($I = 0$). The pipette solution contained (in mM) 138 KCl, 10 NaCl, 1 MgCl$_2$, 10 HEPES, pH 7.2 with KOH (290 mOsm) and freshly added 25 µg/ml gramicidin (Sigma, G5002). Electrical access with a resistance of <50 MΩ was obtained after 5–15 min. Cells were superfused with isotonic solutions containing (in mM) 120 NaCl, 5 KCl, 1 MgCl$_2$, 2.5 CaCl$_2$, 20 HEPES, pH 7.4 and different glucose (1, 5, or 15) concentrations, with osmolarity adjusted to 290 mOsm with mannitol. Hypotonic solutions contained 90 NaCl, 5 KCl, 1 MgCl$_2$, 2.5 CaCl$_2$, 20 HEPES, pH 7.4 and different glucose (1 or 5) concentrations (220 mOsm). All recordings were performed at room temperature. We used an EPC-10 patch-clamp amplifier controlled with the PatchMaster software (HEKA). Data were analyzed using the software pClamp 10.6 (Molecular Devices). Patch pipettes were pulled to a resistance of 2–5 MΩ.

**β-cell mass and pancreatic mass**. β-cell mass was calculated from immunohistochemical stainings as the ratio of insulin-positive area to total pancreatic tissue area, each of which was measured from three different tissue sections per animal (two animals per genotype) using ImageJ and multiplied by the weight of the organ. The relative pancreatic mass was measured as the ratio of the organ weight to body weight.

**Cell volume measurements of β cells**. Cell volume measurements employed the calcein method[70]. Cells were loaded with 4 µM calcein AM (Affymetrix eBiosciences) at 37 °C for 30 min. Live-cell dishes mounted on an Axiovert 200 microscope (Zeiss) were extensively washed with isosmotic solution. Cells were recorded for 2 min in isosmotic solution (in mM: 90 NaCl; 1 MgCl$_2$; 2 CaCl$_2$; 10 glucose; 10 HEPES; 100 mannitol pH 7.4; 310 mOsm before switching to hypotonic solution (210 mOsm), in which mannitol was omitted). For glucose stimulation experiments, cells were superfused and recorded for a minimum of 5 min in 3 mM glucose-containing isosmotic solution (same as used for cytoplasmic Ca$^{2+}$ measurements; 290 mOsm) and then exposed to 25 mM glucose-containing isosmotic solution. Isotonicity of 3 and 25 mM glucose solutions was achieved with mannitol. For recordings, fluorescence was excited at λ = 496 nm while emission signals were recorded at λ = 510 nm (Polychrome V, TILL Photonics) and images acquired at 30 s intervals. Fluorescence of selected ROIs was measured. Data are presented as $F - F_0$ values, where $F_0$ is the fluorescence in isosmotic solution at time 0, and $F$ is the fluorescence at time $t$.

**In vivo experiments**. Adult male mice were fasted for 4 h prior to intraperitoneal injection of glucose (2 mg/g body weight). Blood was withdrawn from the tail and glucose levels were measured with a blood glucose meter (Contour XT, Bayer) at time points 0, 15, 30, 60, and 120 min post injection. Following a 48 h resting period, the same animals used for the GTT were again injected intraperitoneally with insulin (0.75 U/kg body weight) and blood glucose levels were measured as described before. GSIS was performed at the Mouse Metabolic Evaluation Facility (MEF) in Lausanne (Switzerland) according to their experimental design and institutional guidelines. Animals were intraperitoneally injected with glucose (2 mg/g body weight) and blood was withdrawn from the tail vein 60 min before, as well as 2 and 15 min after glucose injection. Plasma insulin levels were evaluated using an insulin-ELISA. In all cases, the genotype of the animals was only revealed after the completion of the experiment.

**Data availability**. The data that support the findings of this study are available from the corresponding author upon reasonable request.

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

## Acknowledgements

We thank Johanna Jedamzick and Carolin Backhaus for technical assistance, Dr. Tobias Stauber for help in generating *Lrrc8a*^lox/lox mice from targeted EUCOMM ES cells, Dr. Fréderic Preitner (Mouse Metabolic Facility, Université de Lausanne) for in vivo insulin secretion experiments, and Dr. Mathew Poy and Dr. Sudhir Tattikota for advice in establishing pancreatic islet purification and insulin secretion measurements. This work was supported by European Research Council (ERC) Advanced Grants (FP7/2007–2013) 294435 "Cytovolion" and (Horizon 2020) 740537 "Volsignal," the Deutsche Forschungsgemeinschaft (Exc257 "Neurocure"), and the Prix Louis-Jeantet de Médecine to T.J.J.

## Author contributions

T.S. designed, performed, and evaluated the majority of the experiments, including western blots, electrophysiology and Ca²⁺ imaging, islet insulin secretion, and in vivo experiments. R.P.-C. designed, generated, and analyzed knock-in mice expressing epitope-tagged LRRC8D, and performed and evaluated cell volume measurements and immunohistochemistry. T.J.J. initiated the study, designed, and evaluated experiments, and wrote the paper. T.S. and R.P.-C. contributed to the writing, prepared figures, and commented on the paper.

## Additional information

**Competing interests:** The authors declare no competing interests.

