## [Peer Review File · Nature Communications]

Reviewers' comments:

Reviewer #1 (Remarks to the Author):

Volume-Regulated Anion Channel (VRAC) may constitute additional KATP-independent glucose-sensing mechanism in pancreatic β -cells. Building on the recent VRAC molecular cloning by Jentsch and Patapoutian groups, Stuhlmann et al tested this important hypothesis and presented a thorough study on the role of LRRC8/VRAC chloride channel in regulating glucose-stimulated pancreatic β -cell excitability and insulin secretion. They found that LRRC8 proteins are expressed in pancreas and islets. By generating a conditional knockout mouse allele, they showed that the essential VRAC subunit LRRC8A is required for hypotonicity- and high glucose-induced VRAC currents in mouse β -cells. They further demonstrated that loss of LRRC8A modestly affects glucose-induced electrical activity, intracellular Ca^{2+} response, and insulin secretion (at 8 min, but not 30 min). The paper is well written and results presented clearly. However, there are two main issues limiting the overall impact and significance of the manuscript.

1. Primary β -cells lacking LRRC8A have minor effects on high glucose-stimulated electric activity and Ca^{2+} influx. Consistently, upon high glucose stimulation, LRRC8A-deficient islets secrete slightly less or similar insulins compared to control islets. Although there is a statistically significant difference in insulin release after 8 min, it's not clear whether it's physiologically significant because the amounts of insulin secreted in the initial phase are very low (less than 0.1%). Thus, among many other cation and chloride channels implicated in KATP-independent β -cell depolarization, VRAC might not be an important/significant modulator of glucose induced insulin secretion in β -cells. The authors didn't observe any gross phenotype in islet morphology and KATP channels are fully functional, however, the delayed Ca^{2+} response could still be due to other general β -cell defects caused by loss of VRAC function.

2. In contrast to the minor insulin secretion defects in vitro, the conditional knockout mice did exhibit an impressive phenotype in glucose tolerance shown in Fig 6c. The GTT assay is the only data supporting a role of LRRC8A in glucose metabolism in vivo. However, it has to be interpreted with caution due to a number of critical issues raised in the field in the last decade on RIP-Cre lines such as Tg(Ins2-cre)^{23Herr} used in this study (See review Magnuson MA et al 2013). For example, RIP-Cre activities were detected across the central nervous system (Wicksteed B et al 2010). More concerning than the leakage of Cre expression, RIP-Cre was constructed with the human growth hormone (hGH) minigene, which was recently reported to affect β -cell functions by itself (Brouwers B et al 2014). It would be more convincing that alternative β -cell specific Cre lines (such as MIP-CreER) are used to examine the physiological role of LRRC8A in insulin-secreting β -cells. Besides GTT, measurements of plasma insulin levels may provide more direct evidence of the involvement of VRAC in glucose-stimulated insulin secretion in vivo.

Minor point: It's assumed and concluded in the abstract that high extracellular glucose causes β -cell volume increase, which in turn activates VRAC. It would be more convincing to show data on glucose-stimulated β -cell volume changes for control and KO β -cells in Figure 3 to support this conclusion.

Reviewer #2 (Remarks to the Author):

Pancreatic beta-cells respond to glucose with calcium influx, a modest swelling and insulin secretion. The glucose-induced beta-cell swelling activates a volume-regulated VRAC anion channel, which has been predicted to play an important role in tuning beta-cell function. The manuscript by Till Stuhlmann et al. details for the first time that LRRC8A is a subunit of the beta-cell VRAC channel. The manuscript determines that glucose increases LRRC8/VRAC activity and that this provides a depolarizing influence on the beta-cell membrane potential that enhances glucose-stimulated calcium entry and insulin secretion. Moreover, the LRRC8/VRAC channel is shown to play an important role in regulating glucose tolerance, thus, beta-cell deficient LRRC8A mice show glucose intolerance. The manuscript includes a significant amount of high quality data relevant to understanding the importance of LRRC8/VRAC channels to beta-cell function. There appears to be a few issues that require attention. These are specified below.

Major:

As the activation of VRAC has important influences on beta cell functions in response to swelling by allowing regulatory volume decrease (RVD), cellular function may become altered over time due to loss of VRAC mediated RVD. This could influence many aspects of beta cell function such as the activities of other ion channels that are sensitive to cell swelling (e.g. KATP and voltage-dependent calcium channels) or it may have negative influences on beta cell health (e.g. changes in inflammasomes activation). Thus, some experiments to ensure that the loss of VRAC is only influencing calcium based on its acute influence on the membrane potential would be important.

The Glucose-activated VRAC current is important; this is the first major current besides KATP that has been identified. As the size of the beta-cell typically adjusts fairly quickly after glucose treatment in part due to VRAC activation, this current would be predicted to be transient as the authors indicate in the discussion. Therefore, it would be good to add some temporal analysis of this VRAC current activation and how long it remains active for following glucose simulation to help increase the understanding of its control of beta cell membrane potential. Moreover, it may be that the

current follows the oscillations in islet swelling thus turning on and off during the oscillatory phases of islet calcium influx. Thus, some analysis of calcium oscillations would add important insights into the importance of this channel for 2nd phase insulin secretion.

Minor:

The following sentence needs references, “. Glucose-sensing by β -cells involves Glut2-mediated cellular uptake of glucose and its conversion to ATP and other metabolites.” While Glut2 is the primary glucose transporter in mouse beta-cells, Glut1 is also present in human beta-cells.

The images in figure 1f show some yellow in the merge images of glucagon and insulin staining. This would indicate that some of the islet cells may be expressing both glucagon and insulin. The amount of these bihormonal cells shown in the β c- Δ 8a mice also seems to be greater. As glucagon and insulin colocalization is not typical in mouse pancreas, either a different imaging technique such as confocal is needed or some more representative sections are required for this figure.

In the Figure 3 legend the following line is presumably a mistake, “2cError! Reference source not found.”

In the discussion the authors mention that, “. Accordingly a marked effect of VRAC on insulin secretion was only observed during the first ~15 minutes of glucose addition to isolated islets.” However, the manuscript shows that insulin secretion differed when looked at for 8 minutes. It is not clear why the authors state 15 minutes when they only tested 8 minutes for the short duration?

Reviewer #3 (Remarks to the Author):

This is an interesting and provocative new paper on the role of VRAC Cl ion channels in beta cell stimulus-secretion coupling. The importance of the work is that it may well represent the molecular identification of previously described volume sensitive Cl currents described in beta cells and provide insights into their possible physiological role.

While the efforts to selectively delete LRRC8 in beta cells in mouse are pretty convincing, the physiological impact is more important to establish, especially given that overall, the mouse lacks a clear phenotype. There are some specific concerns that require attention regarding the manuscript that must be addressed.

MAJOR

1. The data shown in Fig. 1 are generally convincing but the westerns shown in Fig. 1C would benefit from quantitation to firm up the finding of selective knockout and statistics of the result.

2. The data shown in Fig. 2 support the conclusion that hypotonic solution activates a Cl current in WT mouse beta cells but not the deletion mutants. However, as the authors decry the use of drugs such as DCPIB by the field one wonders why they did not also show that siRNA against VRAC in WT blocks the rectifying Cl current?

3. While the authors confirmed that their Cl channel current exhibited the expected anion selectivity expected for canonical VRAC, two other groups did not find this to be the case. Do the authors have an explanation for this discrepancy? Also, might testing a complete series of potentially permeant anions shed additional light?

4. Fig. 3A shows that 20 mM glucose causes activation of the current in WT but not VRAC knockouts. Was cell volume measured in this case? And did application of DCPIB alter volume? As cyclic AMP was also reported to activate islet Cl current, was this also tested?

5. Fig. 5 is especially important to the paper since here, physiological rather than swelling-related stimuli are chosen for study. One can argue that hypotonic solutions are not especially relevant to the study of beta cells as they are not likely to be subjected to external swelling stress in patients with normal kidney function. However, in Fig. 5 glucose is used which is relevant. The issues with this figure, however, are that little is documented short of a time lag between glucose introduction and the activation of spiking or free Ca concentration. What about longer term recordings of spiking or bursting cells or islets? How do the results depend on the concentration of glucose being applied to the islet? Is the dose response right shifted in the knockouts? And for the Ca data in Fig. 5f, the raw tracings should be shown in addition to the plot of the fura2 responses.

6. Fig. 6 has its own issues. First, the choice of glucose challenges is somewhat off as only 3.3 and 25 mM glucose are studied and the latter is a hammer dose. Much nicer if the authors also showed 11-

15 mM doses as in Fig. 5. Also, as they claim an important factor is the timing of the bout of secretion, it would behoove them to do a proper glucose perfusion experiment where flowing solution is continuously collected as it travels over a small number of islets. This would be helpful for documenting the true actions of channel deletion on both phase 1 and phase 2 of secretion.

7. With regards to the impressive data in Fig. 6C, which shows clear glucose intolerance after genetic deletion of LRRC8a, I have two concerns. First, it has been shown that the RIP2 promoter is not specific for only beta cells but can alter metabolism through actions in the hypothalamus. Do the authors have data ruling out expression in hypothalamus? Second, it was not clear whether islets from control animals lacking cre but having floxed LRRC8a or having cre but lacking floxed LRRC8a were all analyzed?

8. Do the authors have data proving that the Cl channel under study is indeed localized only to the beta cell plasma membrane and not to intracellular organelles?

MINOR

1. The comments about % of islet containing beta vs. other cell types included on P. 3 depend on the species being discussed. About 80% of mouse islets are beta cells, while a smaller percentage is present in human. Also, it is more proper to speak not of changes in ATP but changes in ATP/ADP in response to glucose stimulation of beta cells.

2. The sentence on line 75 of P. 4 is incomplete.

Reviewer #4 (Remarks to the Author):

In this paper, Stuhlmann and colleagues show that mouse pancreatic beta cells express the volume-regulated anion channel subunits LRRC8A and LRRC8D. They generated a mouse line deficient of the essential LRRC8A channel subunit and show that beta cells undergo cell swelling and have increased LRRC8A-dependent Cl⁻ current in response to glucose stimulation. The ability of glucose to induce

membrane depolarization, electrical activity and Ca²⁺ influx was delayed in the LRRC8A-deficient beta cells. This effect was specific to glucose and was not observed when beta cells were depolarized using the K-ATP channel blocker tolbutamide or increasing the extracellular K concentration. The delay in membrane depolarization also resulted in reduced initial insulin secretion and glucose tolerance in vivo. The study is well conducted and written and addresses an important aspect of the pancreatic beta cell stimulus-secretion coupling. I have a few concerns with the study that needs to be addressed:

1. The authors state that the *Lrrc8a* deficient mice have a normal lifespan but do not provide any data to support it.
2. I am concerned that the mice were only allowed 48 hr recovery between the GTT and ITT. The animals need a longer recovery period (1 week) between tests. The lack of effect in the ITT is likely correct but I would encourage the authors to repeat the ITT in rested animals.
3. The conclusions that insulin secretion in the *Lrrc8a* deficient mice is reduced in response to glucose, but not tolbutamide and KCl is not supported by the data since all agents were only measured at 30 min and not 8 min where the difference in secretion was observed. Please measure insulin secretion in response to tolbutamide and KCL at 8 min.
4. With the availability of published single human beta cell RNAseq data, it would be interesting if the authors could briefly discuss the translatability of their findings in mice to the human situation.

Detailed responses to the reviewers

We thank all reviewers for appreciating our work and for their insightful comments.

Reviewer #1 (Remarks to the Author):

Volume-Regulated Anion Channel (VRAC) may constitute additional KATP-independent glucose-sensing mechanism in pancreatic β -cells. Building on the recent VRAC molecular cloning by Jentsch and Patapoutian groups, Stuhlmann et al tested this important hypothesis and presented a thorough study on the role of LRRC8/VRAC chloride channel in regulating glucose-stimulated pancreatic β -cell excitability and insulin secretion. They found that LRRC8 proteins are expressed in pancreas and islets. By generating a conditional knockout mouse allele, they showed that the essential VRAC subunit LRRC8A is required for hypotonicity- and high glucose-induced VRAC currents in mouse β -cells. They further demonstrated that loss of LRRC8A modestly affects glucose-induced electrical activity, intracellular Ca^{2+} response, and insulin secretion (at 8 min, but not 30 min). The paper is well written and results presented clearly. However, there are two main issues limiting the overall impact and significance of the manuscript.

We thank the reviewer for appreciating the importance of our paper and stating that it is clearly presented.

1. Primary β -cells lacking LRRC8A have minor effects on high glucose-stimulated electric activity and Ca^{2+} influx. Consistently, upon high glucose stimulation, LRRC8A-deficient islets secrete slightly less or similar insulins compared to control islets. Although there is a statistically significant difference in insulin release after 8 min, it's not clear whether it's physiologically significant because the amounts of insulin secreted in the initial phase are very low (less than 0.1%). Thus, among many other cation and chloride channels implicated in KATP-independent β -cell depolarization, VRAC might not be an important/significant modulator of glucose induced insulin secretion in β -cells. The authors didn't observe any gross phenotype in islet morphology and KATP channels are fully functional, however, the delayed Ca^{2+} response could still be due to other general β -cell defects caused by loss of VRAC function.

We agree with the reviewer that the effect of VRAC deletion on β -cell excitability and insulin secretion is less than that observed with deletion/inhibition of KATP channels. Clearly VRAC is not required for β -cell excitability or Ca^{2+} transients since the 'canonical' KATP pathway remains intact. An only modulatory effect is therefore expected, and the onset of VRAC-dependent cell volume regulation should temporally limit the effect of VRAC as stated in the manuscript.

We agree with the reviewer that one can never totally exclude secondary effects of a gene KO. However, the unchanged morphology, normal insulin contents of KO islets, intact KATP response, plus our new data showing that we do not detect markers of inflammation (experiments done in

response to reviewer 2) do not provide indications that secondary changes cause the delayed response to glucose.

Most importantly, our paper shows the presence of swelling-activated VRAC currents in WT, but not in Lrrc8a KO β -cells, the induction of Lrrc8a-dependent VRAC currents by high glucose (which we now show directly to induce swelling), and the Lrrc8a (VRAC)- dependent glucose-induced depolarization in β -cells. These data very strongly support a direct role of VRAC currents in the observed effects on β -cell excitability and insulin secretion.

In order to further support such a direct role of VRAC currents in β -cell excitability, we also tested inhibitors (carbenoxolone and DCPIB) on the Ca^{2+} responses of β -cells to glucose (Fig. R1 for the reviewers). Carbenoxolone strongly inhibited glucose-induced, but not tolbutamide-induced Ca^{2+} transients in isolated WT β -cells in roughly 90% of the cells. However, this inhibition was stronger than observed in our Lrrc8a KO cells. The remaining 10% of the cells showed a reduction in Ca^{2+} response that resembled those of the Lrrc8a KO cells. The crux here is the lack of specific inhibitors. Carbenoxolone also inhibits e.g. connexin and pannexins hemichannels and possibly other channels. Similar problems are encountered with DCPIB, the 'most specific' VRAC inhibitor, which abolished not only the response to glucose, but also to tolbutamide, possibly by activating KATP channels (Best...Nilius, Eur J Pharmacol. 2014). While the carbenoxolone experiments are compatible with the proposed role of VRAC, they can certainly not prove it, stressing even more the need of clean genetic KOs as used in our paper.

Although we are convinced that our conclusion on the role of VRAC in insulin secretion is very well supported by the experimental evidence, we inserted a half sentence as a caveat in the last sentence of Results: 'Although, like in every gene disruption experiment, we cannot totally exclude secondary effects, we conclude that LRRC8/VRAC channels of β -cells modulate the secretion of insulin and the regulation of blood glucose.'

2. In contrast to the minor insulin secretion defects in vitro, the conditional knockout mice did exhibit an impressive phenotype in glucose tolerance shown in Fig 6c. The GTT assay is the only data supporting a role of LRRC8A in glucose metabolism in vivo. However, it has to be interpreted with caution due to a number of critical issues raised in the field in the last decade on RIP-Cre lines such as Tg(Ins2-cre)23Herr used in this study (See review Magnuson MA et al 2013). For example, RIP-Cre activities were detected across the central nervous system (Wicksteed B et al 2010). More concerning than the leakage of Cre expression, RIP-Cre was constructed with the human growth hormone (hGH) minigene, which was recently reported to affect β -cell functions by itself (Brouwers B et al 2014). It would be more convincing that alternative β -cell specific Cre lines (such as MIP-CreER) are used to examine the physiological role of LRRC8A in insulin-secreting β -cells

Indeed, the RIP-Cre line, which we and many other have used to efficiently delete floxed genes in β -cells, has been described to express Cre-recombinase also in a small number of CNS neurons. We have re-assessed this issue and have crossed the RIP-Cre mice to tdTomato reporter mice to examine the expression of Cre in the CNS (Fig. R2 for the reviewers). We indeed see a sparse

population of Cre-positive cells scattered over the brain, including the hypothalamus, as described previously. However, we believe that this expression in the brain is not relevant for the present study. In particular, our experiments on isolated β -cells, which provide very strong evidence for the proposed signal transduction cascade, would be very difficult to explain by an effect of the CNS. While it might have been better to use another Cre-line from the beginning, we cannot repeat nearly all experiments with another Cre-line. This would not only require more than a year of work (and a lot of research money), but would also be impossible to justify to the authorities to whom we would have to apply for a new animal experiment permit.

While the sparse brain expression of our Cre-line seems to be a rather minor concern of the reviewer, he/she is more concerned about the possible effect of Cre-expression per se in β -cells (as described by others). We have performed the appropriate control in our in vivo glucose- and insulin tolerance experiments. There is no discernible difference between the $Lrrc8a^{+/+}; Cre^+$ and $Lrrc8a^{lox/lox}; Cre-$ groups. These data, which indicate that expression of the Cre-recombinase in β -cells does not cause glucose intolerance, have now been added to Fig. 6c, d.

Besides GTT, measurements of plasma insulin levels may provide more direct evidence of the involvement of VRAC in glucose-stimulated insulin secretion in vivo.

We agree with the reviewer that the demonstration of an involvement of VRAC in glucose-stimulated insulin secretion in vivo would further support our conclusions. Unfortunately, it is impossible to get the necessary permit for these new animal experiments from the Berlin authorities in less than 6-8 months. We therefore contacted the Mouse Metabolic Evaluation Facility (MEF) of the Université de Lausanne, Switzerland, who have extensive experience in these assays, and asked them to perform these experiments on a paid service basis.

Indeed, the new experiments, which are shown in the new Fig. 6e, suggest that VRAC is important for insulin secretion in vivo. Whereas we observe a nearly two-fold increase in insulin plasma levels of WT mice 2 min after glucose injection, such an increase is absent in beta-cell specific LRRC8A KO mice, with insulin levels being similar again at 15 min. Although with the number of mice tested (13 - 14 per genotype), t-tests give a marginal level of significance ($p=0.07$), we think that these data are a valuable addition to our paper and support the role of VRAC in early insulin secretion.

Minor point: It's assumed and concluded in the abstract that high extracellular glucose causes β -cell volume increase, which in turn activates VRAC. It would be more convincing to show data on glucose-stimulated β -cell volume changes for control and KO β -cells in Figure 3 to support this conclusion.

As mentioned in our paper, β -cell swelling by high extracellular glucose has been published by another group previously (Miley et al., J. Physiol. 1997). Nonetheless, following the suggestion of the reviewer, we have now measured swelling of β -cells in response to glucose which we show as

new panel a in Fig. 3. The extent of this swelling is minor compared to swelling induced by the very drastic, non-physiological decrease in extracellular osmolarity, which is used by almost everybody in the field to induce VRAC activity. For comparison, we now also show hypotonicity-induced β -cell swelling in Fig. 2a. See also our response to point 4 of reviewer 3.

Reviewer #2 (Remarks to the Author):

Pancreatic beta-cells respond to glucose with calcium influx, a modest swelling and insulin secretion. The glucose-induced beta-cell swelling activates a volume-regulated VRAC anion channel, which has been predicted to play an important role in tuning beta-cell function. The manuscript by Till Stuhlmann et al. details for the first time that LRRC8A is a subunit of the beta-cell VRAC channel. The manuscript determines that glucose increases LRRC8/VRAC activity and that this provides a depolarizing influence on the beta-cell membrane potential that enhances glucose-stimulated calcium entry and insulin secretion. Moreover, the LRRC8/VRAC channel is shown to play an important role in regulating glucose tolerance, thus, beta-cell deficient LRRC8A mice show glucose intolerance. The manuscript includes a significant amount of high quality data relevant to understanding the importance of LRRC8/VRAC channels to beta-cell function. There appears to be a few issues that require attention. These are specified below.

We thank the reviewer for appreciating the high quality and importance of our paper.

Major:

As the activation of VRAC has important influences on beta cell functions in response to swelling by allowing regulatory volume decrease (RVD), cellular function may become altered over time due to loss of VRAC mediated RVD. This could influence many aspects of beta cell function such as the activities of other ion channels that are sensitive to cell swelling (e.g. KATP and voltage-dependent calcium channels) or it may have negative influences on beta cell health (e.g. changes in inflammasomes activation). Thus, some experiments to ensure that the loss of VRAC is only influencing calcium based on its acute influence on the membrane potential would be important.

This point, very similar to point 1 of reviewer 1, is well taken. For a detailed response, please see our answer to reviewer 1. In short, we did not detect changes in morphology, insulin contents, and KATP channel activity, and the demonstrated effects on a cellular level (e.g. activation of VRAC by glucose, VRAC-induced depolarization....) are totally consistent with a direct effect of VRAC currents (or their lack) on insulin secretion.

Following the advice of the reviewer, we now have also investigated whether we detect signs of inflammation in KO islets. We did not detect fibrosis in KO islets (not shown), and

Immunohistochemistry did not reveal differences in active caspase 3 or NLRP3 (an inflammasome component) between WT and KO islets. These new data are now displayed as new Suppl. Fig. 2. As stated in the response to reviewer 1, we also performed new experiments using acute VRAC inhibition with carbenoxolone. While the resulting data (Fig. R1 for the reviewers) are compatible with a direct role of VRAC, the lacking specificity of this (and any other known) inhibitor of VRAC limits the conclusions that can be drawn from this experiment. It rather emphasizes the need for a clean gene disruption as done in the present study.

The Glucose-activated VRAC current is important; this is the first major current besides KATP that has been identified. As the size of the beta-cell typically adjusts fairly quickly after glucose treatment in part due to VRAC activation, this current would be predicted to be transient as the authors indicate in the discussion. Therefore, it would be good to add some temporal analysis of this VRAC current activation and how long it remains active for following glucose stimulation to help increase the understanding of its control of beta cell membrane potential. Moreover, it may be that the current follows the oscillations in islet swelling thus turning on and off during the oscillatory phases of islet calcium influx. Thus, some analysis of calcium oscillations would add important insights into the importance of this channel for 2nd phase insulin secretion.

We agree with the reviewer that the VRAC current should be transient, as we had already discussed in the manuscript. We have quite extensive experience with VRAC currents measured in HEK cells for our structure-function studies. In these whole-cell experiments, in which we stimulate VRAC with commonly used rather brutal decreases in extracellular osmolarity, VRAC currents reach a maximum after about 8-17 minutes and then begin to decline with variable kinetics. This decline could be to various factors, including some intrinsic channel inactivation and changes in cytosolic ion composition. While we agree with the reviewer that it may be interesting to know the time-course of the decline of glucose-induced VRAC currents in β -cells, these experiments are exceedingly difficult and prone to artifacts. In our hands, it is very rare to get stable perforated patch recordings of β -cells for more than 15-20 minutes, and we assume that even perforated patch measurements will affect β -cells and VRAC inactivation over that time period. Therefore, we were unable to perform the suggested experiments.

However, following the reviewer, we did measure intracellular calcium oscillations of intact WT and KO islets over longer periods of time. In both genotypes, oscillations vary quite a bit between individual islets, showing difference oscillation patterns as described previously. However, there was no obvious difference in oscillations between the genotypes. We show these data in the new Supplementary Fig. 4 and in Fig. R3 for the reviewers. We refer to these results on pages 7-8 in the main text.

Minor:

The following sentence needs references, “. Glucose-sensing by β -cells involves Glut2-mediated cellular uptake of glucose and its conversion to ATP and other metabolites.” While Glut2 is the primary glucose transporter in mouse beta-cells, Glut1 is also present in human beta-cells.

We had indeed focused here on Glut2 because our study is concerned with mice. In order to not discuss the presence of different Glut isoforms in different species, which may disturb the flow of the text, we have changed the wording to a more generic: ‘Glucose-sensing by β -cells involves glucose transporter-mediated cellular uptake of glucose.’

The images in figure 1f show some yellow in the merge images of glucagon and insulin staining. This would indicate that some of the islet cells may be expressing both glucagon and insulin. The amount of these bihormonal cells shown in the β c- Δ 8a mice also seems to be greater. As glucagon and insulin colocalization is not typical in mouse pancreas, either a different imaging technique such as confocal is needed or some more representative sections are required for this figure.

We have investigated this issue in many new IHC experiments, with several pictures shown for the reviewers in Fig. R4. We see no significant overlap between insulin and glucagon labeling in either genotype and have replaced the image in Fig. 1f by a new one to avoid confusion. (Please note that printing on a color printer at this resolution can still artificially lead to yellow color).

In the Figure 3 legend the following line is presumably a mistake, “2cError! Reference source not found.”

Thank you, yes, that was a mistake introduced by EndNote, which has now been corrected.

In the discussion the authors mention that, “. Accordingly a marked effect of VRAC on insulin secretion was only observed during the first ~15 minutes of glucose addition to isolated islets.” However, the manuscript shows that insulin secretion differed when looked at for 8 minutes. It is not clear why the authors state 15 minutes when they only tested 8 minutes for the short duration?

Thank you for pointing this out. This statement (~15 min) was based on our belief that a marked difference would last longer than the eight minutes, the time at which we did the measurement, since there was still a (non-significant) tendency of lower insulin secretion at 30 min (Fig. 6a). Of course, you are right that this is just an assumption not backed up by hard data. Therefore, we have corrected the sentence to: ‘. Accordingly a marked effect of VRAC on insulin secretion was only observed at 8 minutes of glucose addition to isolated islets.’

Reviewer #3 (Remarks to the Author):

This is an interesting and provocative new paper on the role of VRAC Cl ion channels in beta cell stimulus-secretion coupling. The importance of the work is that it may well represent the molecular identification of previously described volume sensitive Cl currents described in beta cells and provide insights into their possible physiological role.

We thank the reviewer for appreciating the importance of our paper.

While the efforts to selectively delete LRRC8 in beta cells in mouse are pretty convincing, the physiological impact is more important to establish, especially given that overall, the mouse lacks a clear phenotype. There are some specific concerns that require attention regarding the manuscript that must be addressed.

MAJOR

1. The data shown in Fig. 1 are generally convincing but the westerns shown in Fig. 1C would benefit from quantitation to firm up the finding of selective knockout and statistics of the result.

We have now quantified the data from three Western blots of lysates from three different mouse pairs. We display the results in a new bar diagram in Fig. 1c which shows a marked, highly significant reduction in Lrrc8a expression in beta cell KO islets.

2. The data shown in Fig. 2 support the conclusion that hypotonic solution activates a Cl current in WT mouse beta cells but not the deletion mutants. However, as the authors decry the use of drugs such as DCPIB by the field one wonders why they did not also show that siRNA against VRAC in WT blocks the rectifying Cl current?

We totally agree with the reviewer that our genetic KO is much preferable to inhibitors such as DCPIB, since all known VRAC inhibitors are non-specific. We included the DCPIB inhibition to connect to the previous literature which used DCPIB and other inhibitors to identify VRAC currents. Using genomic disruption of Lrrc8a, we now show that the previously described DCPIB-sensitive volume-activated current is indeed mediated by LRRC8/VRAC channels. The suggested use of Lrrc8a siRNA would be inferior to our genomic Lrrc8a disruption since siRNAs are not 100% efficient and may have off-target effects.

3. While the authors confirmed that their Cl channel current exhibited the expected anion selectivity expected for canonical VRAC, two other groups did not find this to be the case. Do the authors have an explanation for this discrepancy? Also, might testing a complete series of potentially permeant anions shed additional light?

We do not have an explanation for the fact that two groups reported a Cl>I selectivity sequence for β -cell VRAC instead of the typical I>Cl selectivity of that current. Our study now shows that β -cell VRAC has indeed, as expected, the I>Cl selectivity which has been published for VRAC in any

of the tissues and cells where this has been examined. Moreover, this selectivity fits 100% to LIRC8 channels, which we now show to mediate these currents. In fact, we have shown in Voss et al (Science 2014) that combinations of Lrrc8a with any of the other Lrrc8 isoforms yield currents with this selectivity sequence. Testing extended series of other anions has not been done for the tested β -cell lines previously, and we would consider this beyond the scope of our manuscript.

We can only speculate that the Cl^{>I} selectivity reported for β -cell lines resulted from a contamination with background currents, or other experimental problems.

4. Fig. 3A shows that 20 mM glucose causes activation of the current in WT but not VRAC knockouts. Was cell volume measured in this case? And did application of DCPIB alter volume? As cyclic AMP was also reported to activate islet Cl current, was this also tested?

We have now followed this reviewer (and reviewer 1 (minor point)) and have measured the swelling of β -cells in response to extracellular glucose (25 mM) using the calcein method (shown in new panel a of Fig. 3). Even with 25 mM glucose, this cell swelling is significantly less than elicited with a strong decrease in extracellular osmolarity, which we now show for comparison in Fig. 2a. Importantly, as we now also state in the text, both WT and Lrrc8a KO β -cells swell to similar degrees, excluding impaired glucose-induced cell swelling as mechanism for reduced β -cell excitability and insulin secretion with Lrrc8a KO. Interestingly, while regulatory volume decrease (RVD) is clearly visible in the continued presence of hypotonicity (Fig. 2a), there is no clear sign of RVD during the same time span with high glucose. This may suggest that the continued metabolism of glucose compensates during that time for the release of osmolytes through VRAC, as now hypothesized in Discussion (page 10-11).

We did not test the effect of DCPIB on cell volume, because the beta-cell specific Lrrc8a KO is the most specific way to test for an involvement of VRAC, and because DCPIB is notoriously non-specific. The latter point is bolstered by our new experiments in which we tested DCPIB on glucose-induced Ca²⁺ oscillations (Fig. R1 for reviewers), where DCPIB also abolished the response of KATP. The effect of cAMP has previously been attributed to an effect on CFTR (Guo et al, Nat Comm. 2014) and was not investigated further in our study.

5. Fig. 5 is especially important to the paper since here, physiological rather than swelling-related stimuli are chosen for study. One can argue that hypotonic solutions are not especially relevant to the study of beta cells as they are not likely to be subjected to external swelling stress in patients with normal kidney function. However, in Fig. 5 glucose is used which is relevant. The issues with this figure, however, are that little is documented short of a time lag between glucose introduction and the activation of spiking or free Ca concentration.

What about longer term recordings of spiking or bursting cells or islets?

As stated above (response to reviewer 2) it is difficult to keep stable perforated patches over a long period of time, and even this technique, though more gentle than whole-cell recordings, is

likely to affect the cell (e.g. by cytosolic changes in the concentration of cations, which permeate through gramicidin). We therefore preferred to address this issue with measurements of intracellular calcium, as requested also by reviewer 2. We now show our evaluation of cytosolic Ca^{2+} oscillations in WT and beta-cell specific *Lrrc8a* KO islets in the new Supplementary Fig. 4 and in Figure R3 for the reviewers. No significant difference between the genotypes could be detected.

How do the results depend on the concentration of glucose being applied to the islet? Is the dose response right shifted in the knockouts?

*This is indeed an interesting question. We now performed measurements of cytosolic Ca-concentrations of WT and KO β -cells in response to 6 and 8 mM glucose (see Suppl. Figure 3). Already at 6 mM glucose, there may be a tendency of KO β -cells to respond in a delayed manner to glucose. This delay, which is very clear at 15 mM glucose, becomes more apparent at 8 mM glucose. We feel that the data are not precise enough to deduce dose-response curves here. When we refer to these new and interesting data, to be on the safe side, we state: 'A tendency for a delayed glucose-response of *Lrrc8a*^{-/-} β -cells became already apparent in the 6 to 8 mM glucose concentration range (Supplementary Fig. 3)'*

And for the Ca data in Fig. 5f, the raw tracings should be shown in addition to the plot of the fura2 responses.

We agree that it is useful to show the primary raw traces. We have now added these both to Fig. 5f and to the new Suppl. Fig. 3 described above.

6. Fig. 6 has its own issues. First, the choice of glucose challenges is somewhat off as only 3.3 and 25 mM glucose are studied and the latter is a hammer dose. Much nicer if the authors also showed 11-15 mM doses as in Fig. 5. Also, as they claim an important factor is the timing of the bout of secretion, it would behoove them to do a proper glucose perfusion experiment where flowing solution is continuously collected as it travels over a small number of islets. This would be helpful for documenting the true actions of channel deletion on both phase 1 and phase 2 of secretion.

We agree, of course, that 11-15 mM external glucose is more physiological than the 'hammer dose' of 25 mM we have used previously (although this concentration can be nearly reached in glucose tolerance measurements, see Fig. 6). We have now performed new experiments using only 10 mM glucose, and indeed also observe a significant, roughly 50% decrease of insulin secretion (with sampling for 8 min), very similar to the results obtained previously for 25 mM. We have now shown the response to 10 mM glucose in the main text (Fig. 6b) and have moved the data for 25 mM glucose into the Supplement (Suppl. Fig. 5).

7. With regards to the impressive data in Fig. 6C, which shows clear glucose intolerance after genetic

deletion of LRRC8a, I have two concerns. First, it has been shown that the RIP2 promoter is not specific for only beta cells but can alter metabolism through actions in the hypothalamus. Do the authors have data ruling out expression in hypothalamus? Second, it was not clear whether islets from control animals lacking cre but having floxed LRRC8a or having cre but lacking floxed LRRC8a were all analyzed?

Please see our detailed response to reviewer 1 who had the same concern in his/her point 2. In short, we have confirmed that the used Cre line is expressed in scarce, scattered cells of the brain, including the hypothalamus (Figure for reviewers R2). However, the insulin tolerance test shows that there is no effect on peripheral glucose metabolism, it seems unlikely that it influences the acute glucose tolerance experiment, and an influence on our in vitro experiments with isolated islets or β -cells is almost excluded. Moreover, we have now included the Cre+, loxP negative controls that yield glucose- or insulin- tolerance results indistinguishable from Cre-, loxP controls (see Fig. 6c, d).

8. Do the authors have data proving that the Cl channel under study is indeed localized only to the beta cell plasma membrane and not to intracellular organelles?

This is a good point – indeed, immunohistochemistry for our Lrrc8d-tdTomato mice (Fig. 1b) seems to indicate ‘cytoplasmic’ (i.e. probably vesicular) labeling for tdTomato. For many epithelial cells (unpublished), however, we see clear plasma membrane labeling. At this point, we cannot exclude the possibility that the fusion of the rather large tdTomato to the Lrrc8d C-terminus influences the trafficking or even folding of Lrrc8d. In this case, the ‘cytoplasmic’ localization seen in Fig. 1b might be an artifact. It should also be noted that LRRC8D, when not associated with LRRC8A in heterologous overexpression, is stuck in the ER (Voss et al, Science 2014). Since readers may have the same question, we have added the sentence: ‘Immunolabeling of LRRC8D-tdTomato stained not only the plasma membrane, but also prominently the cytoplasm of β -cells (Fig. 1b). However, we cannot be sure whether this represents a significant presence of native LRRC8D-containing VRACs in intracellular organelles, or may have rather been caused by the fusion of the large epitope to the LRRC8D carboxy-terminus.’

MINOR

1. The comments about % of islet containing beta vs. other cell types included on P. 3 depend on the species being discussed. About 80% of mouse islets are beta cells, while a smaller percentage is present in human. Also, it is more proper to speak not of changes in ATP but changes in ATP/ADP in response to glucose stimulation of beta cells.

*We have now cited a review of Rorsman (Prog Biophys Mol Biol **107**, 224-235 (2011)) who gives a value of 75% for the proportion of beta-cells in rodent islets (Page 5) and mention also at other instances the 75% value.*

Further, we have twice replaced ATP by ATP/ADP ratio, which is certainly more correct.

2. The sentence on line 75 of P. 4 is incomplete.

This sentence 'β-cell swelling induced by hypotonicity or glucose activated canonical, LRRC8A-dependent $I_{Cl,vol}$ currents which depolarized the cell.' was, in principle, complete, but could be rather easily misread. We have now bracketed 'induced by hypotonicity or glucose' with commas to avoid this misunderstanding.

Reviewer #4 (Remarks to the Author):

In this paper, Stuhlmann and colleagues show that mouse pancreatic beta cells express the volume-regulated anion channel subunits LRRC8A and LRRC8D. They generated a mouse line deficient of the essential LRRC8A channel subunit and show that beta cells undergo cell swelling and have increased LRRC8A-dependent Cl⁻ current in response to glucose stimulation. The ability of glucose to induce membrane depolarization, electrical activity and Ca²⁺ influx was delayed in the LRRC8A-deficient beta cells. This effect was specific to glucose and was not observed when beta cells were depolarized using the K-ATP channel blocker tolbutamide or increasing the extracellular K concentration. The delay in membrane depolarization also resulted in reduced initial insulin secretion and glucose tolerance in vivo. The study is well conducted and written and addresses an important aspect of the pancreatic beta cell stimulus-secretion coupling.

We thank the reviewer for appreciating the importance and quality of our work.

I have a few concerns with the study that needs to be addressed:

1. The authors state that the Lrrc8a deficient mice have a normal lifespan but do not provide any data to support it.

Thank you for pointing this out. Our statement was imprecise. We did not perform a systematic study of lifespan, waiting for a year or more for mice to die of natural causes. We rather meant that there was no increased lethality of beta-cell specific KO mice until 30 weeks of age, an age after which all mice had been sacrificed for experiments. We have now added new data (in Fig. 1j) showing that there was no difference in weight development during this time between the genotypes.

We now state at the end of the third paragraph of results: 'βc-Δ8a mice showed no increased lethality during our observation period of 30 weeks and their weight developed normally during that time (Fig. 1j). The wording (at the end of the first paragraph of results) has been replaced by: '...were viable and lacked an overt phenotype.'

2. I am concerned that the mice were only allowed 48 hr recovery between the GTT and ITT. The animals need a longer recovery period (1 week) between tests. The lack of effect in the ITT is likely correct but I would encourage the authors to repeat the ITT in rested animals.

Unfortunately, we cannot repeat these experiments. This would require a new animal experiment permit from the authorities, which most likely would not be granted because it would be a repetition of a previously performed experiment. In any case, obtaining such a permit would take more than 6 months in Berlin. It should be noted, however, that initial blood glucose levels from GTT, and ITT experiments are indistinguishable between the genotypes. Moreover, as stated in response to reviewer 1, we have performed (with an external metabolic facility) new glucose-induced insulin secretion experiments in vivo (new Fig. 6e). The difference in insulin secretion between the genotypes cannot be explained by changed peripheral glucose uptake.

3. The conclusions that insulin secretion in the *Lrrc8a* deficient mice is reduced in response to glucose, but not tolbutamide and KCl is not supported by the data since all agents were only measured at 30 min and not 8 min where the difference in secretion was observed. Please measure insulin secretion in response to tolbutamide and KCL at 8 min.

We have now done the suggested experiments. These are shown in the new panel b of Fig. 6 together with insulin secretion in response to the more physiological stimulus of 10 mM glucose. These new data support our conclusions.

4. With the availability of published single human beta cell RNAseq data, it would be interesting if the authors could briefly discuss the translatability of their findings in mice to the human situation.

*Thank you for this suggestion. The work of Segerstolpe... Sandberg, Single cell transcriptome profiling of human pancreatic islets..., Cell Metabolism 2016, finds all *LRRC8* subunits in pancreatic alpha, beta and gamma cells. They find rather low expression of *LRRC8C* and *LRRC8E* in single cell sequencing, which may agree to the low abundance of *LRRC8E* in our study (but less so with *LRRC8C*). However, these data are difficult to compare, as our data depend on the affinity of the antibody used. However, it might be possible to compare the relative abundance in endocrine vs. exocrine pancreas. For *LRRC8E* the data of Segerstolpe et al. are compatible with ours (less presence of *LRRC8E* in endocrine than in exocrine pancreas both in mice and humans), whereas the higher abundance of *LRRC8C* in endocrine than exocrine pancreas, as observed in our study for mice, is not reflected in the Segerstolpe data for human pancreas.*

*Following the reviewer, we added the following sentence to our manuscript on page 5: 'For comparison, human pancreatic islets express all *LRRC8* subunits as determined by single-cell transcriptome profiling³². Although these data are difficult to compare, human *LRRC8E* appears less expressed in islets compared to exocrine pancreas as found here for mice, but human *LRRC8C**

or -D expression does not seem to differ greatly between islets and acinar/ductal cells³².’ (with ref³² referring to Segerstolpe et al.)

Data extracted from Table S1 of Segerstolpe et al. (the higher the number, the lower the abundance):

	α-cells	β-cells	γ-cells	acinar cells	Ductal cells
L8A	8435	9364	9209	4978	4819
L8B	10163	9008	9958	8550	8420
L8C	16125	15095	14373	14717	15221
L8D	5393	5090	6909	6166	6459
L8E	16960	16983	18341	10056	8637
Out of 26000					

Figure R1. Blockade of glucose-stimulated currents by DCPIB and Carbenoxolone (Cbx). 20 μ M DCPIB or 100 μ M Cbx were added 5 min before the experiment. Cells were loaded with Fura-2 in bath solution containing 3 mM glucose. During the experiment, cells were kept (with or without inhibitor) for 1 min in 3 mM glucose solution and afterwards for 19 min in 15 mM glucose solution, followed by 10 min in 300 μ M tolbutamide + 15 mM glucose solution. Cells from 2 animals were tested for each experiment. At least 13 cells were measured per condition.

Figure R2. *Ins2-Cre* mediated recombinase activity is scarce and not restricted to specific regions of the brain hypothalamus.

Ins2-Cre mice were bred to TdTomato reporter mouse line (B6.Cg-*Gt(ROSA)26Sor^{tm9(CAG-tdTomato)Hze/J}*). Images of two individual coronal brain slices show an anterior (A) and a posterior (B) view of the hypothalamus. 50 μm thick free floating slices were stained with RFP antibody and nuclei counterstained with DAPI. Tomato staining is shown in gray while merged images show nuclei co-staining. Zoom images in left column display detailed staining in different brain areas.

3V, third ventricle; PVA= Paraventricular thalamic nucleus; DM= Dorsomedial hypothalamic nuclei

Figure R3. Calcium oscillations in intact islets. Different oscillation patterns were observed for both genotypes after stimulation of intact islets with 10 mM glucose. 10 mM glucose was already added 30 min before measuring. N=18 for *Lrrc8a*^{lox/lox} and β_c - $\Delta 8a$ islets from 4 different animals each.

a

b

Figure R4. Normal number and distribution of β - and α -cells in pancreatic islets upon β -cell-specific *Lrrc8a* disruption. Immunofluorescent staining of pancreatic islet sections of (a) *Lrrc8a*^{lox/lox} and (b) β c- Δ 8a mice using β - and α -cell marker proteins insulin (green) and glucagon (red), respectively. Representative islets from two separate mice per genotype; mouse ID number followed by genotype is depicted in the left.

REVIEWERS' COMMENTS:

Reviewer #1 (Remarks to the Author):

Till Stuhlmann et al. provided some discussion and additional data on the two major limitations raised last time. While some questions were addressed, there are still remaining concerns on those two major points.

1. The modest modulatory role of LRRC8A in regulating glucose-stimulated insulin secretion, Ca²⁺ influx and electrical activity (Figure 5-6) is in direct contrast with a recent paper published in Nature Communications by Rajan Sah and colleagues (SWELL1 is a glucose sensor regulating β -cell excitability and systemic glycaemia). Data presented in that study showed that SWELL1/LRRC8A is absolutely required for glucose-stimulated insulin secretion, Ca²⁺ influx and electrical activity in both primary mouse β -cells and primary human β -cells. The authors should cite and discuss the differences. Because they are the main findings of two otherwise very similar studies, it's thus important to understand and resolve where the major differences came from. Rajan Sah et al. acutely knocked out SWELL1/LRRC8A through Cre expression in flox/flox mouse β -cells. They also acutely knocked down SWELL1 through shRNA expression in human β -cells. However, Till Stuhlmann et al. isolated β -cells from control and conditional knockout mice, where LRRC8A was deleted early on during development. It is known that many other ion channels, both cation (TRP) and chloride channels (TMEM16, CFTR etc), can contribute to β -cell excitability. Is it possible that other ionic mechanisms partially compensate the loss of LRRC8A during development? This can be readily tested with Cre expression in isolated flox/flox mouse β -cells. Any other potential explanation for this major discrepancy and confusion?

2. The authors added Ins-Cre mice as an important control and showed they behave similarly as flox/flox mice, which excluded the effect of transgene expression. However, consistent to the literature, the authors did find scattered yet widespread Ins-Cre mediated recombinase activities all cross the brain (I encourage the authors show the brain cre expression data together with those of pancreas in Figure S6). I disagree with the authors' believe that leaky Cre expression "is not relevant for the present study". Deletion of VRAC in those neuronal populations may partially contribute to the impaired glucose tolerance through well-established central regulation of energy metabolism, as it is revealed for some Ins-Cre conditional ko mice in recent years. This is the reason that β -cell-specific inducible Cre (like MIP-CreER) is a preferred tool in order to draw a convincing conclusion. The point is well taken on the constraint with new cre line breeding. However, the in vitro data cannot exclude the brain contribution to the glucose tolerance phenotype in vivo, which is critical to support the physiological importance of VRAC in β -cells claimed in the study.

Reviewer #2 (Remarks to the Author):

The manuscript has been improved with the additional data and revisions, which have addressed all of my previous concerns. The manuscript uncovers an important role for the LRRC8/VRAC in modulating glucose-stimulated beta-cell calcium influx and insulin secretion.

Reviewer #3 (Remarks to the Author):

The authors have done a good job responding to the critiques.

Reviewer #4 (Remarks to the Author):

No further comments - nice work!

Detailed response to the reviewers of Stuhlmann, Planells-Cases and Jentsch (2nd review)

*We thank all four reviewers for taking the time and effort to review our revised manuscript. Whereas reviewers 2, 3 and 4 were fully satisfied with the quite extensive additions and changes incorporated into our revised manuscript (see below), reviewer 1 raised a new point based on a paper by Sah and colleagues that had appeared in the meantime, and repeated his/her criticism concerning the *Ins2Cre* line we have used.*

Detailed responses:

Reviewer #1 (Remarks to the Author):

Till Stuhlmann et al. provided some discussion and additional data on the two major limitations raised last time. While some questions were addressed, there are still remaining concerns on those two major points.

1. The modest modulatory role of LRRC8A in regulating glucose-stimulated insulin secretion, Ca²⁺ influx and electrical activity (Figure 5-6) is in direct contrast with a recent paper published in Nature Communications by Rajan Sah and colleagues (SWELL1 is a glucose sensor regulating β -cell excitability and systemic glycaemia). Data presented in that study showed that SWELL1/LRRC8A is absolutely required for glucose-stimulated insulin secretion, Ca²⁺ influx and electrical activity in both primary mouse β -cells and primary human β -cells. The authors should cite and discuss the differences. Because they are the main findings of two otherwise very similar studies, it's thus important to understand and resolve where the major differences came from. Rajan Sah et al. acutely knocked out SWELL1/LRRC8A through Cre expression in flox/flox mouse β -cells. They also acutely knocked down SWELL1 through shRNA expression in human β -cells. However, Till Stuhlmann et al. isolated β -cells from control and conditional knockout mice, where LRRC8A was deleted early on during development. It is known that many other ion channels, both cation (TRP) and chloride channels (TMEM16, CFTR etc), can contribute to β -cell excitability. Is it possible that other ionic mechanisms partially compensate the loss of LRRC8A during development? This can be readily tested with Cre expression in isolated flox/flox mouse β -cells. Any other potential explanation for this major discrepancy and confusion?

*When we saw the paper of Sah and coworkers, which came out during the review process (our PubMed search for VRAC or LRRC8 did not find it), we were very surprised that the authors assert that *Lrrc8a* (VRAC) is absolutely required for glucose-induced insulin secretion and cytoplasmic Ca²⁺ transients. This assertion is truly unexpected in view of the major role of *K_{ATP}* in glucose sensing (the 'canonical pathway'), which should remain unaffected in *Lrrc8a*^{-/-} β -cells. Moreover, a total lack of glucose-induced insulin secretion is very difficult to reconcile with the rather mild phenotype of β -cell specific *Lrrc8a* KO mice generated either in Sah's or our lab, which both have, for instance, normal glucose serum levels under resting conditions.*

*It appears that also the reviewers of the Sah paper (as gleaned from their comments available at the Nature Communications website) were astonished by that finding, as it seems to imply that VRAC is upstream of *K_{ATP}* signaling, with this pathway being somehow inhibited in the absence of VRAC (reviewer 1). The same point was raised by their reviewer 2, who brought up this point again in the second round of*

comments and suggested, as we do, that K_{ATP} inhibition and VRAC activation are rather additive and cooperative in glucose-stimulated excitation of beta cells.

Following a request of the reviewers, Kang, Sah and coworkers stimulated β -cells with different concentrations of glibenclamide, a K_{ATP} blocker. They observed (their Fig. 6 i, l) that $Lrrc8a^{-/-}$ β -cells need higher concentrations of glibenclamide to elicit Ca^{2+} -transients, showing that (1) K_{ATP} signaling is functional and (2) that VRAC sensitizes the parallel K_{ATP} pathway, as suggested by us.

The hypothesis that VRAC is constitutively open under resting conditions to such a degree that its deletion, by abolishing depolarizing Cl^- currents, would hyperpolarize the cell and thereby prevent an effect of K_{ATP} closure on V is not feasible. First, VRAC is almost completely closed under isotonic conditions (our Fig. 2d,e). Second, neither we (our Fig. 4a) nor Kang et al. (their Fig. 3 b,e,h) found a difference in β -cell resting V between WT and KO.

But how to explain the seemingly complete absence of glucose-induced Ca -transients in murine and human β -cells lacking VRAC, or expressing less VRAC (shRNA) (Fig. 5b, f of Sah and coworkers)? Of course, we don't really know because we did not perform those experiments.

One point might be the rather short time of glucose stimulation by Kang et al.. We found that VRAC enhances the sensitivity of glucose sensing, yielding a shorter delay until electrical activity (our Fig. 5a-c) or Ca increases (our Fig. 5f,g) are elicited. We followed this time course for up to 15 min, whereas Sah and colleagues used 5-8 minutes. We strongly assume, also based on their results with K_{ATP} inhibitors (see above), that they would have seen Ca -transients with longer exposure to glucose (which indirectly inhibits K_{ATP}) also in VRAC KO cells. There are also some differences in experimental conditions (such as buffer compositions), but these seem unlikely to have affected the results.

As pointed out by the reviewer, a significant difference between the two studies is that we used cells from β -cell specific KO mice, whereas Kang et al. eliminated or suppressed $Lrrc8a$ more acutely using adenoviral transduction with vectors expressing Cre-recombinase or shRNA, respectively. One may certainly argue, as the reviewer did, that in the KO mouse model the β -cell may have compensated the loss of VRAC by changed expression of other channels (we checked for K_{ATP} and found it unchanged, but of course we cannot exclude changes in other channels or pathways). This could have led to a more moderate effect in our case. On the other hand, also acute overexpression of Cre-recombinase could have had effects, off-target effects of the shRNA cannot be excluded, and the loss of VRAC during a few days (as in the transduction experiment) may have effects on other channels as well. All of this is speculative at this point.

The reviewer suggested that we essentially repeat in our lab the experiments of Sah and co-workers, i.e. rapidly (within a few days) delete $Lrrc8a$ with transduction of Cre-recombinase in $Lrrc8a^{lox/lox}$ β -cells. This would put the burden to explain the difference on our, rather than Sah's lab. These experiments might confirm or falsify the 'compensation' hypothesis, but would take a lot of time and effort – to breed again enough mice, newly establish the appropriate vectors and adenoviral transduction (which we have not used before). These experiments will take at least 4 additional months for us to perform, and, by further delaying the publication of our paper, it would considerably increase the difference in publication dates of both papers, which would not only be bad for us, but also for the readers. We therefore prefer not to do this experimental comparison for the present manuscript.

To summarize and to be frank, we feel that the claim of Kang et al. that *Lrrc8a* is REQUIRED for glucose-stimulated Ca transients and insulin secretion is exaggerated and that Kang et al. would have seen a response if they would have stimulated KO cells with glucose for a longer time. Our results showing that VRAC modulates insulin secretion, but is not required for this process, are compatible with what is known about the major role of K_{ATP} and insulin secretion. In contrast, an absolute requirement of VRAC for insulin secretion would be very surprising and difficult to reconcile with other findings such as the intactness of signal transduction by K_{ATP} and the mild phenotype of KO mice. We also believe that publication in Nature Communications of both our and Sah's papers, ideally accompanied with a News and Views type comment comparing our findings, would well serve the scientific community.

CHANGES MADE:

We now discuss the recent paper from the Sah lab, and the apparent discrepancies to our work, in a new paragraph in Discussion:

'While this manuscript was under review, Sah and coworkers reported⁵⁰ that the LRRC8A subunit is a 'glucose sensor' required for glucose-induced excitation and insulin secretion of β -cells. Both studies coincide in that VRAC plays a role in glucose-sensitivity of β -cells *in vitro* and *in vivo* and agree in several other aspects. However, in stark contrast to our study, Kang et al. reported that glucose-induced Ca^{2+} -transients and insulin secretion were abolished in *Lrrc8a*^{-/-} β -cells, rather than only being delayed or decreased as found by us. The suggested absolute requirement for VRAC⁵⁰ is surprising in view of the rather mild phenotypes of β -cell-specific *Lrrc8a*^{-/-} mice and unchanged resting glucose levels in either study, and also because the major, canonical K_{ATP} pathway for glucose sensing remains intact upon *Lrrc8a* disruption. Moreover, consistent with VRAC being closed at rest, both studies reported that the resting potential of *Lrrc8a*^{-/-} β -cells is unchanged. This virtually eliminates the possibility that β -cell hyperpolarization counteracts the depolarizing effect of K_{ATP} closure in *Lrrc8a*^{-/-} mice. A possible explanation for the discrepancy might be owed to methodological differences. Whereas we used β -cells from $\beta c-\Delta 8a$ mice, Kang et al.⁵⁰ more acutely disrupted *Lrrc8a* with adenoviral transduction of Cre-recombinase into *Lrrc8a*^{lox/lox} cells, or alternatively reduced *Lrrc8a* expression by transduction of shRNA. Whereas in our study the lack of VRAC might have been compensated by altered expression of other channels (although the K_{ATP} pathway appeared unchanged), the acute viral overexpression of either Cre-recombinase or shRNA by Kang et al.⁵⁰ might have caused secondary changes that further decreased the glucose-sensitivity of *Lrrc8a*^{-/-} β -cells. Another decisive factor may be the rather short exposure of β -cells to glucose by Kang et al.⁵⁰, who thus may have missed the delayed glucose-response of KO cells found here.'

2. The authors added Ins-Cre mice as an important control and showed they behave similarly as flox/flox mice, which excluded the effect of transgene expression. However, consistent to the literature, the authors did find scattered yet widespread Ins-Cre mediated recombinase activities all cross the brain (I encourage the authors show the brain cre expression data together with those of pancreas in Figure S6). I disagree with the authors' believe that leaky Cre expression "is not relevant for the present study". Deletion of VRAC in those neuronal populations may partially contribute to the impaired glucose

tolerance through well-established central regulation of energy metabolism, as it is revealed for some Ins-Cre conditional ko mice in recent years. This is the reason that β -cell-specific inducible Cre (like MIP-CreER) is a preferred tool in order to draw a convincing conclusion. The point is well taken on the constraint with new cre line breeding. However, the in vitro data cannot exclude the brain contribution to the glucose tolerance phenotype in vivo, which is critical to support the physiological importance of VRAC in β -cells claimed in the study.

We agree with the reviewer that we cannot totally exclude an effect of the deletion of VRAC in some of the CNS neurons on in vivo insulin secretion, and that our cursory statement that this expression pattern is irrelevant was not justified. On the other hand, Kang, Sah et al. used the inducible β -cell specific Cre line which is considered to be the best suited by the field (and the reviewer) and observed in essence the same in vivo phenotype. Taking this new paper into account, we therefore conclude that putative effects of VRAC deletion in the CNS do not significantly affect our results.

CHANGES MADE:

We now discuss this issue in the Methods section and, as requested by the reviewer, have added a new panel that shows Cre-expression in the brain (using a cross with reporter mice) to Supplementary Figure 6.

The addition to Methods reads:

*'Although the Ins2-Cre (*Tg(Ins2-cre)23Herr*) mouse expresses the recombinase also in sparse cells scattered over the brain^{67,68}, which we confirmed by crossing this strain with reporter mice (Supplementary Fig. 6b), this is not a concern since Kang et al.⁵⁰ used the recommended *Ins1Cre^{ERT2}* strain to delete *Lrrc8a* and obtained results *in vivo* similar to ours.'*

Reviewer #2 (Remarks to the Author):

The manuscript has been improved with the additional data and revisions, which have addressed all of my previous concerns. The manuscript uncovers an important role for the LRRC8/VRAC in modulating glucose-stimulated beta-cell calcium influx and insulin secretion.

Reviewer #3 (Remarks to the Author):

The authors have done a good job responding to the critiques.

Reviewer #4 (Remarks to the Author):

No further comments - nice work!

We thank again all reviewers for their insightful comments, which led to a significant improvement of our manuscript.